# Exemplary Natural Images Explain CNN Activations Better than State-of-the-Art Feature Visualization

**Judy Borowski**[*], **Roland S. Zimmermann**[*], **Judith Schepers, Robert Geirhos, Thomas S. A. Wallis**[†‡]**, Matthias Bethge**[‡]**, Wieland Brendel**[‡]
University of Tübingen, Germany

## Abstract

Feature visualizations such as synthetic maximally activating images are a widely used explanation method to better understand the information processing of convolutional neural networks (CNNs). At the same time, there are concerns that these visualizations might not accurately represent CNNs' inner workings. Here, we measure how much extremely activating images help humans to predict CNN activations. Using a well-controlled psychophysical paradigm, we compare the informativeness of synthetic images by Olah et al. (2017) with a simple baseline visualization, namely exemplary natural images that also strongly activate a specific feature map. Given either synthetic or natural reference images, human participants choose which of two query images leads to strong positive activation. The experiment is designed to maximize participants' performance, and is the first to probe *intermediate* instead of final layer representations. We find that synthetic images indeed provide helpful information about feature map activations ($82 \pm 4\%$ accuracy; chance would be $50\%$). However, natural images — originally intended to be a baseline — outperform these synthetic images by a wide margin ($92 \pm 2\%$). Additionally, participants are faster and more confident for natural images, whereas subjective impressions about the interpretability of the feature visualizations by Olah et al. (2017) are mixed. The higher informativeness of natural images holds across most layers, for both expert and lay participants as well as for hand- and randomly-picked feature visualizations. Even if only a single reference image is given, synthetic images provide less information than natural images ($65 \pm 5\%$ vs. $73 \pm 4\%$). In summary, synthetic images from a popular feature visualization method are significantly less informative for assessing CNN activations than natural images. We argue that visualization methods should improve over this simple baseline.

## 1 Introduction

As Deep Learning methods are being deployed across society, academia and industry, the need to understand their decisions becomes ever more pressing. Under certain conditions, a "right to explanation" is even required by law in the European Union (GDPR, 2016; Goodman & Flaxman, 2017). Fortunately, the field of *interpretability* or *explainable artificial intelligence* (XAI) is also growing: Not only are discussions on goals and definitions of interpretability advancing (Doshi-Velez & Kim, 2017; Lipton, 2018; Gilpin et al., 2018; Murdoch et al., 2019; Miller, 2019; Samek et al., 2020) but the number of explanation methods is rising, their maturity is evolving (Zeiler & Fergus, 2014; Ribeiro et al., 2016; Selvaraju et al., 2017; Kim et al., 2018) and they are tested and

---

[*]Joint first and corresponding authors: `firstname.lastname@uni-tuebingen.de`
[†]Current affiliation: Institute of Psychology and Center for Cognitive Science, Technische Universität Darmstadt
[‡]Joint senior authors

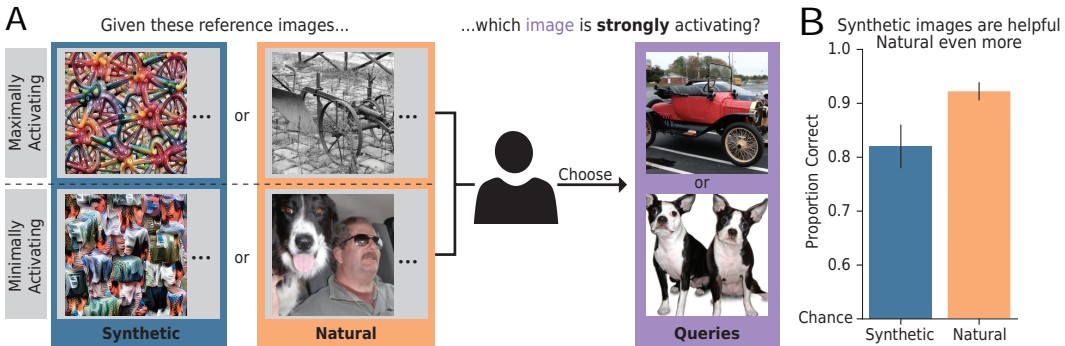

Figure 1: How useful are synthetic compared to natural images for interpreting neural network activations? **A: Human experiment.** Given extremely activating reference images (either synthetic or natural), a human participant chooses which out of two query images is also a strongly activating image. Synthetic images were generated via feature visualization (Olah et al., 2017). **B: Core result.** Participants are well above chance for synthetic images — but even better when seeing *natural* reference images.

used in real-world scenarios like medicine (Cai et al., 2019; Kröll et al., 2020) and meteorology (Ebert-Uphoff & Hilburn, 2020).

We here focus on the popular post-hoc explanation method (or interpretability method) of *feature visualizations via activation maximization*[1]. First introduced by Erhan et al. (2009) and subsequently improved by many others (Mahendran & Vedaldi, 2015; Nguyen et al., 2015; Mordvintsev et al., 2015; Nguyen et al., 2016a; 2017), these synthetic, maximally activating images seek to visualize features that a specific network unit, feature map or a combination thereof is selective for. However, feature visualizations are surrounded by a great controversy: How accurately do they represent a CNN's inner workings—or in short, how useful are they? This is the guiding question of our study.

On the one hand, many researchers are convinced that feature visualizations are interpretable (Graetz, 2019) and that "features can be rigorously studied and understood" (Olah et al., 2020b). Also other applications from Computer Vision and Natural Language Processing support the view that features are meaningful (Mikolov et al., 2013; Karpathy et al., 2015; Radford et al., 2017; Zhou et al., 2014; Bau et al., 2017; 2020) and might be formed in a hierarchical fashion (LeCun et al., 2015; Güçlü & van Gerven, 2015; Goodfellow et al., 2016). Over the past few years, extensive investigations to better understand CNNs are based on feature visualizations (Olah et al., 2020b;a; Cammarata et al., 2020; Cadena et al., 2018), and the technique is being combined with other explanation methods (Olah et al., 2018; Carter et al., 2019; Addepalli et al., 2020; Hohman et al., 2019).

On the other hand, feature visualizations can be equal parts art and engineering as they are science: vanilla methods look noisy, thus human-defined regularization mechanisms are introduced. But do the resulting beautiful visualizations accurately show what a CNN is selective for? How representative are the seemingly well-interpretable, "hand-picked" (Olah et al., 2017) synthetic images in publications for the entirety of all units in a network, a concern raised by e.g. Kriegeskorte (2015)? What if the features that a CNN is truly sensitive to are imperceptible instead, as might be suggested by the existence of adversarial examples (Szegedy et al., 2013; Ilyas et al., 2019)? Morcos et al. (2018) even suggest that units of easily understandable features play a less important role in a network. Another criticism of synthetic maximally activating images is that they only visualize extreme features, while potentially leaving other features undetected that only elicit e.g. 70% of the maximal activation. Also, polysemantic units (Olah et al., 2020b), i.e. units that are highly activated by different semantic concepts, as well as the importance of combinations of units (Olah et al., 2017; 2018; Fong & Vedaldi, 2018) already hint at the complexity of how concepts are encoded in CNNs.

One way to advance this debate is to measure the utility of feature visualizations in terms of their helpfulness for *humans*. In this study, we therefore design well-controlled psychophysical experiments that aim to quantify the informativeness of the popular visualization method by Olah et al. (2017). Specifically, participants choose which of two natural images would elicit a higher activa-

---

[1]Also known as *input maximization* or *maximally exciting images (MEIs)*.

tion in a CNN given a set of reference images that visualize the network selectivities. We use natural query images because real-world applications of XAI require understanding model decisions to natural inputs. To the best of our knowledge, our study is the first to probe how well humans can predict *intermediate* CNN activations. Our data shows that:

- Synthetic images provide humans with helpful information about feature map activations.

- Exemplary natural images are even more helpful.

- The superiority of natural images mostly holds across the network and various conditions.

- Subjective impressions of the interpretability of the synthetic visualizations vary greatly between participants.

## 2 RELATED WORK

Significant progress has been made in recent years towards understanding CNNs for image data. Here, we mention a few selected methods as examples of the plethora of approaches for understanding CNN decision-making: *Saliency maps* show the importance of each pixel to the classification decision (Springenberg et al., 2014; Bach et al., 2015; Smilkov et al., 2017; Zintgraf et al., 2017), *concept activation vectors* show a model's sensitivity to human-defined concepts (Kim et al., 2018), and other methods - amongst feature visualizations - focus on explaining individual units (Bau et al., 2020). Some tools integrate interactive, software-like aspects (Hohman et al., 2019; Wang et al., 2020; Carter et al., 2019; Collaris & van Wijk, 2020; OpenAI, 2020), combine more than one explanation method (Shi et al., 2020; Addepalli et al., 2020) or make progress towards automated explanation methods (Lapuschkin et al., 2019; Ghorbani et al., 2019). As overviews, we recommend Gilpin et al. (2018); Zhang & Zhu (2018); Montavon et al. (2018) and Carvalho et al. (2019).

Despite their great insights, challenges for explanation methods remain. Oftentimes, these techniques are criticized as being over-engineered; regarding feature visualizations, this concerns the loss function and techniques to make the synthetic images look interpretable (Nguyen et al., 2017). Another critique is that interpretability research is not sufficiently tested against falsifiable hypotheses and rather relies too much on intuition (Leavitt & Morcos, 2020).

In order to further advance XAI, scientists advocate different directions. Besides the focus on developing additional methods, some researchers (e.g. Olah et al. (2020b)) promote the "natural science" approach, i.e. studying a neural network extensively and making empirical claims until falsification. Yet another direction is to quantitatively evaluate explanation methods. So far, only decision-level explanation methods have been studied in this regard. Quantitative evaluations can either be realized with humans directly or with mathematically-grounded models as an approximation for human perception. Many of the latter approaches show great insights (e.g. Hooker et al. (2019); Nguyen & Martínez (2020); Fel & Vigouroux (2020); Lin et al. (2020); Tritscher et al. (2020); Tjoa & Guan (2020)). However, a recent study demonstrates that metrics of the explanation quality computed without human judgment are inconclusive and do not correspond to the *human* rankings (Biessmann & Refiano, 2019). Additionally, Miller (2019) emphasizes that XAI should build on existing research in philosophy, cognitive science and social psychology.

The body of literature on human evaluations of explanation methods is growing: Various combinations of data types (tabular, text, static images), task set-ups and participant pools (experts vs. laypeople, on-site vs. crowd-sourcing) are being explored. However, these studies all aim to investigate final model decisions and do not probe intermediate activations like our experiments do. For a detailed table of related studies, see Appendix Sec. A.3. A commonly employed task paradigm is the "forward simulation / prediction" task, first introduced by Doshi-Velez & Kim (2017): Participants guess the model's computation based on an input and an explanation. As there is no absolute metric for the goodness of explanation methods (yet), comparisons are always performed within studies, typically against baselines. The same holds for additional data collected for confidence or trust ratings. According to the current literature, studies reporting positive effects of explanations (e.g. Kumarakulasinghe et al. (2020)) slightly outweigh those reporting inconclusive (e.g. Alufaisan et al. (2020); Chu et al. (2020)) or even negative effects (e.g. Shen & Huang (2020)).

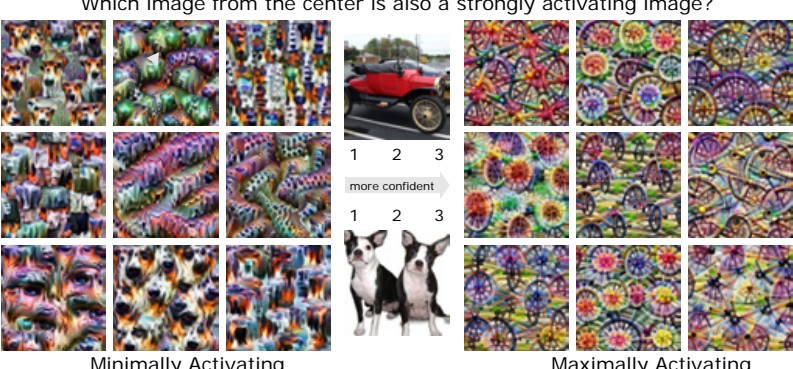

Figure 2: Example trial in psychophysical experiments. A participant is shown minimally and maximally activating reference images for a certain feature map on the sides and is asked to select the image from the center that also strongly activates that feature map. The answer is given by clicking on the number according to the participant's confidence level (1: not confident, 2: somewhat confident, 3: very confident). After each trial, the participant receives feedback which image was indeed the maximally activating one. For screenshots of each step in the task, see Appendix Fig. 7.

To our knowledge, no study has yet evaluated the popular explanation method of feature visualizations and how it improves human understanding of intermediate network activations. This study therefore closes an important gap: By presenting data for a forward prediction task of a CNN, we provide a quantitative estimate of the informativeness of maximally activating images generated with the method of Olah et al. (2017). Furthermore, our experiments are unique as they probe for the first time how well humans can predict *intermediate* model activations.

## 3    METHODS

We perform two human psychophysical studies[2] with different foci (Experiment I ($N = 10$) and Experiment II ($N = 23$)). In both studies, the task is to choose the one image out of two natural query images (two-alternative forced choice paradigm) that the participant considers to also elicit a strong activation given some reference images (see Fig. 2). Apart from the image choice, we record the participant's confidence level and reaction time. Specifically, responses are given by clicking on the confidence levels belonging to either query image. In order to gain insights into how intuitive participants find feature visualizations, their subjective judgments are collected in a separate task and a dynamic conversation after the experiment (for details, see Appendix Sec. A.1.1 and Appendix Sec. A.2.6).

All design choices are made with two main goals: (1) allowing participants to achieve the *best performance possible* to approximate an upper bound on the helpfulness of the explanation method, and (2) gaining a *general* impression of the helpfulness of the examined method. As an example, we choose the natural query images from among those of lowest and highest activations ($\rightarrow$ best possible performance) and test many different feature maps across the network ($\rightarrow$ generality). For more details on the human experiment besides the ones below, see Appendix Sec. A.1.

In Experiment I, we focus on comparing the performance of synthetic images to two baseline conditions: natural reference images and no reference images. In Experiment II, we compare lay vs. expert participants as well as different presentation schemes of reference images. Expert participants qualify by being familiar or having practical experience with feature visualization techniques or at least CNNs. Regarding presentation schemes, we vary whether only maximally or both maximally and minimally activating images are shown; as well as how many example images of each of these are presented (1 or 9).

Following the existing work on feature visualization (Olah et al., 2017; 2018; 2020b;a), we use an Inception V1 network[3] (Szegedy et al., 2015) trained on ImageNet (Deng et al., 2009; Russakovsky

---

[2]Code and data is available at `https://bethgelab.github.io/testing_visualizations/`
[3]also known as GoogLeNet

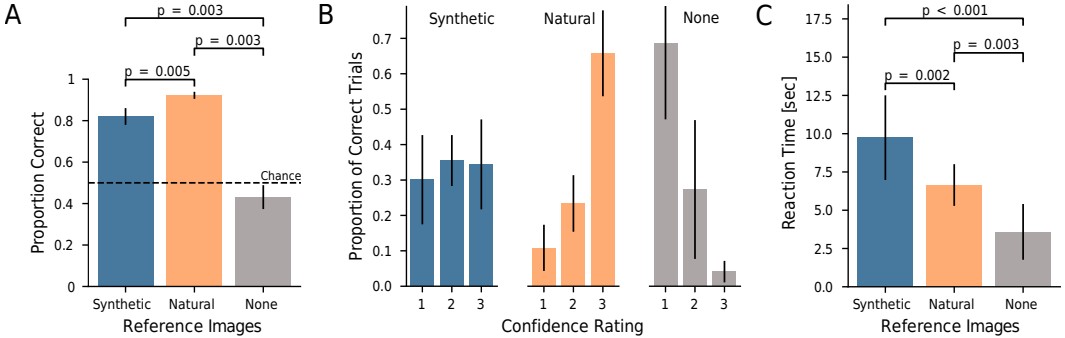

Figure 3: Participants are better, more confident and faster at judging which of two query images causes higher feature map activation with natural than with synthetic reference images. **A: Performance.** Given synthetic reference images, participants are well above chance (proportion correct: $82 \pm 4\%$), but even better for natural reference images ($92 \pm 2\%$). Without reference images (baseline comparison "None"), participants are close to chance. **B: Confidence.** Participants are much more confident (higher rating = more confident) for natural than for synthetic images on correctly answered trials ($\chi^2$, $p < .001$). **C: Reaction time.** For correctly answered trials, participants are on average faster when presented with natural than with synthetic reference images. We show additional plots on confidence and reaction time for incorrectly answered trials and all trials in the Appendix (Fig. 16); for Experiment II, see Fig. 17.). The $p$-values in A and C correspond to Wilcoxon signed-rank tests.

et al., 2015). The synthetic images throughout this study are the optimization results of the feature visualization method by Olah et al. (2017) with the spatial average of a whole feature map ("channel objective"). The natural stimuli are selected from the validation set of the ImageNet ILSVRC 2012 dataset (Russakovsky et al., 2015) according to their activations for the feature map of interest. Specifically, the images of the most extreme activations are sampled, while ensuring that each lay or expert participant sees different query and reference images. A more detailed description of the specific sampling process for natural stimuli and the generation process of synthetic stimuli is given in Sec. A.1.2.

## 4    RESULTS

In this section, all figures show data from Experiment I except for Fig. 5A+C, which show data from Experiment II. All figures for Experiment II, which replicate the findings of Experiment I, as well as additional figures for Experiment I (such as a by-feature-map analysis), can be found in the Appendix Sec. A.2. Note that (unless explicitly noted otherwise), error bars denote two standard errors of the mean of the participant average metric.

### 4.1    PARTICIPANTS ARE BETTER, MORE CONFIDENT AND FASTER WITH NATURAL IMAGES

Synthetic images can be helpful: Given synthetic reference images generated via feature visualization (Olah et al., 2017), participants are able to predict whether a certain network feature map prefers one over the other query image with an accuracy of $82 \pm 4\%$, which is well above chance level ($50\%$) (see Fig. 3A). However, performance is even higher in what we intended to be the baseline condition: natural reference images ($92 \pm 2\%$). Additionally, for correct answers, participants much more frequently report being highly certain on natural relative to synthetic trials (see Fig. 3B), and their average reaction time is approximately 3.7 seconds faster when seeing natural than synthetic reference images (see Fig. 3C). Taken together, these findings indicate that in our setup, participants are not just better overall, but also more confident and substantially faster on natural images.

### 4.2    NATURAL IMAGES ARE MORE HELPFUL ACROSS A BROAD RANGE OF LAYERS

Next, we take a more fine-grained look at performance across different layers and branches of the Inception modules (see Fig. 4). Generally, feature map visualizations from lower layers show low-level features such as striped patterns, color or texture, whereas feature map visualizations from

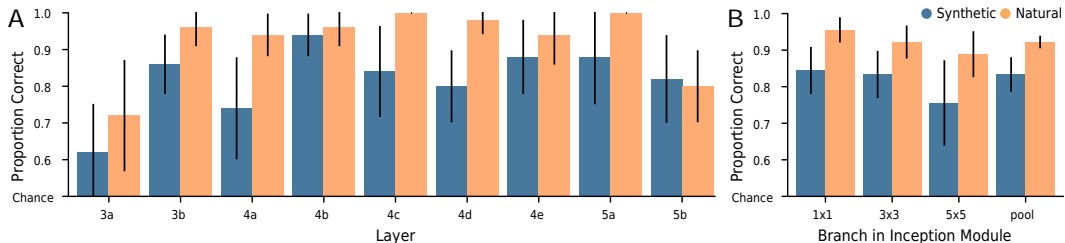

Figure 4: Performance is high across (**A**) a broad range of layers and (**B**) all branches of the Inception modules. The latter differ in their kernel sizes ($1 \times 1$, $3 \times 3$, $5 \times 5$, pool). Again, natural images are (mostly) more helpful than synthetic images. Additional plots for the none condition as well as Experiment II can be found in the Appendix in respectively Fig. 18 and Fig. 19.

higher layers tend to show more high-level concepts like (parts of) objects (LeCun et al., 2015; Güçlü & van Gerven, 2015; Goodfellow et al., 2016). We find performance to be reasonably high across most layers and branches: participants are able to match both low-level and high-level patterns (despite not being explicitly instructed what layer a feature map belonged to). Again, natural images are mostly more helpful than synthetic images.

### 4.3 FOR EXPERT AND LAY PARTICIPANTS ALIKE: NATURAL IMAGES ARE MORE HELPFUL

Explanation methods seek to explain aspects of algorithmic decision-making. Importantly, an explanation should not just be amenable to experts but to anyone affected by an algorithm's decision. We here test whether the explanation method of feature visualization is equally applicable to expert and lay participants (see Fig. 5A). Contrary to our prior expectation, we find no significant differences in expert vs. lay performance (RM ANOVA, $p = .44$, for details see Appendix Sec. A.2.2). Hence, extensive experience with CNNs is not necessary to perform well in this forward simulation task. In line with the previous main finding, both experts and lay participants are both better in the natural than in the synthetic condition.

### 4.4 EVEN FOR HAND-PICKED FEATURE VISUALIZATIONS, PERFORMANCE IS HIGHER ON NATURAL IMAGES

Often, explanation methods are presented using carefully selected network units, raising the question whether author-chosen units are representative for the interpretability method as a whole. Olah et al. (2017) identify a number of particularly interpretable feature maps in Inception V1 in their appendix overview. When presenting either these hand-picked visualizations[4] or randomly selected ones, performance for hand-picked feature maps improves slightly (Fig. 5B); however this performance difference is small and not significant for both natural (Wilcoxon test, $p = .59$) and synthetic (Wilcoxon test, $p = .18$) reference images (see Appendix Sec. A.2.4 for further analysis). Consistent with the findings reported above, performance is higher for natural than for synthetic reference images *even on carefully selected hand-picked feature maps*.

### 4.5 ADDITIONAL INFORMATION BOOSTS PERFORMANCE, ESPECIALLY FOR NATURAL IMAGES

Publications on feature visualizations vary in terms of how optimized images are presented: Often, a single maximally activating image is shown (e.g. Erhan et al. (2009); Carter et al. (2019); Olah et al. (2018)); sometimes a few images are shown simultaneously (e.g. Yosinski et al. (2015); Nguyen et al. (2016b)), and on occasion both maximally and minimally activating images are shown in unison (Olah et al. (2017)). Naturally, the question arises as to what influence (if any) these choices have, and whether there is an optimal way of presenting extremely activating images. For this reason, we systematically compare approaches along two dimensions: the number of reference images (1 vs. 9) and the availability of minimally activating images (only Max vs. Min+Max). The results can

---

[4]All our hand-picked feature maps are taken from the pooling branch of the Inception module. As the appendix overview in Olah et al. (2017) does not contain one feature map for each of these, *we* select interpretable feature maps for the missing layers mixed5a and mixed5b ourselves.

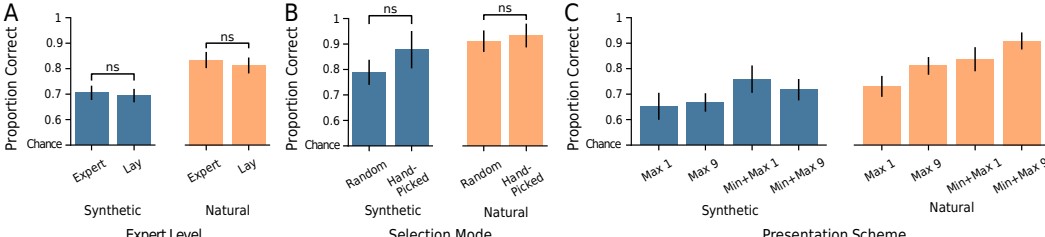

Figure 5: We found no evidence for large effects of expert level or feature map selection. However, performance does improve with additional information. **A: Expert level.** Both experts and lay participants perform equally well (RM ANOVA, $p = .44$), and consistently better on natural than on synthetic images. **B: Selection mode.** There is no significant performance difference between hand-picked feature maps selected for interpretability and randomly selected ones (Wilcoxon test, $p = .18$ for synthetic and $p = .59$ for natural reference images). **C: Presentation scheme.** Presenting both maximally and minimally activating images simultaneously (Min+Max) and presenting nine instead of one single reference image tend to improve performance, especially for natural reference images. "ns" highlights non-significant differences.

be found in Fig. 5C. When just a single maximally activating image is presented (condition Max 1), natural images already outperform synthetic images ($73 \pm 4\%$ vs. $64 \pm 5\%$). With additional information along either dimension, performance improves both for natural as well as for synthetic images. The stronger boost in performance, however, is observed for natural reference images. In fact, performance is higher for natural than for synthetic reference images in all four conditions. In the Min+Max 9 condition, a replication of the result from Experiment I shown in Fig. 3A, natural images now outperform synthetic images by an even larger margin ($91 \pm 3$ vs $72 \pm 4\%$).

### 4.6 SUBJECTIVELY, INTERPRETABILITY OF FEATURE VISUALIZATIONS VARIES GREATLY

While our data suggests that feature visualizations are indeed helpful for humans to predict CNN activations, we want to emphasize again that our design choices aim at an upper bound on their informativeness. Another important aspect of evaluating an explanation method is the subjective impression. Besides recording confidence ratings and reaction times, we collect judgments on *intuitiveness trials* (see Appendix Fig. 14) and oral impressions after the experiments. The former ask for ratings of how intuitive feature visualizations appear for natural images. As Fig. 6A+B show, participants perceive the intuitiveness of synthetic feature visualizations for strongly activating natural dataset images very differently. Further, the comparison of intuitiveness judgments before and after the main experiments reveals only a small significant average improvement for one out of three feature maps (see Fig. 6B+C, Wilcoxon test, $p < .001$ for mixed4b). The interactive conversations paint a similar picture: Some synthetic feature visualizations are perceived as intuitive while others do not correspond to understandable concepts. Nonetheless, four participants report that their first "gut feeling" for interpreting these reference images (as one participant phrased it) is more reliable. Further, a few participants point out that the synthetic visualizations are exhausting to understand. Finally, three participants additionally emphasize that the minimally activating reference images played an important role in their decision-making.

In a by-feature-map analysis (see Appendix A.2.7 for details and images, as well as Supplementary Material 1 for more images), we compare differences and commonalities for feature maps of different performance levels. According to our observations, easy feature maps seem to contain clear object parts or shapes. In contrast, difficult feature maps seem to have diverse reference images, features that do not correspond to human concepts, or contain conflicting information as to which commonalities between query and reference images matter more. Bluntly speaking, we are also often surprised that participants identified the correct image — the reasons for this are unclear to us.

## 5 DISCUSSION & CONCLUSION

Feature visualizations such as synthetic maximally activating images are a widely used explanation method, but it is unclear whether they indeed help humans to understand CNNs. Using well-controlled psychophysical experiments with both expert and lay participants, we here conduct the

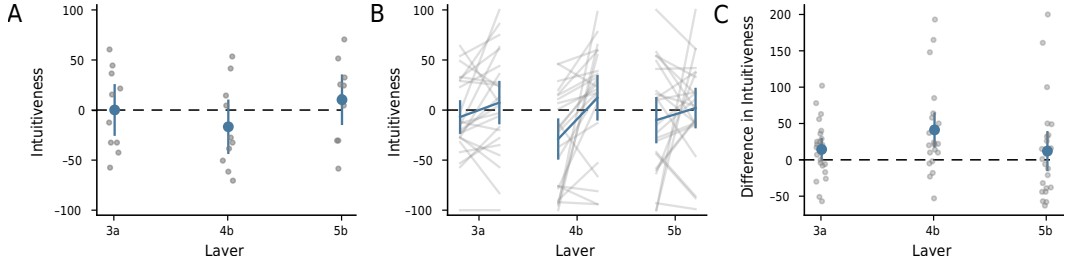

Figure 6: The subjective intuitiveness of feature visualizations varies greatly (see **A** for the ratings from the beginning of Experiment I and **B** for the ratings at the beginning and end of Experiment II). The means over all participants yield a neutral result, i.e. the visualizations are neither un- nor intuitive, and the improvement of subjective intuitiveness before and after the experiment is only significant for one feature map (mixed4b). **C:** On average, participants found feature visualizations slightly more intuitive after doing the experiment as the differences larger than zero show. In all three subfigures, gray dots and lines show data per participant.

very first investigation of intermediate synthetic feature visualizations by Olah et al. (2017): Can participants predict which of two query images leads to a strong activation in a feature map, given extremely activating visualizations? Specifically, we shed light on the following questions:

(1.) *How informative are synthetic feature visualizations — and how do they compare to a natural image baseline?* We find above-chance performance given synthetic feature visualizations, but to our own surprise, synthetic feature visualizations are systematically *less* informative than the simple baseline of strongly activating natural images. Interestingly, many synthetic feature visualizations contain regularization mechanisms to introduce more "natural structure" (Olah et al., 2017), sometimes even called a "natural image prior" (Mahendran & Vedaldi, 2015; Offert & Bell, 2020). This raises the question: Are natural images maybe all you need? One might posit that extremely activating natural (reference) images would have an unfair advantage because we also test on extremely activating natural (query) images. However, our task design ultimately reflects that XAI is mainly concerned with explaining how units behave on *natural* inputs. Furthermore, the fact that feature visualization are not bound to the natural image manifold is often claimed as an advantage because it supposedly allows them to capture more precisely which features a unit is sensitive to (Olah et al., 2017). Our results, though, demonstrate that this is not the case if we want to understand the behavior of units on natural inputs.

(2.) *Do you need to be a CNN expert in order to understand feature visualizations?* To the best of our knowledge, our study is the first to compare the performances of expert and lay people when evaluating explanation methods. Previously, publications either focused on only expert groups (Hase & Bansal, 2020; Kumarakulasinghe et al., 2020) or only laypeople (Schmidt & Biessmann, 2019; Alufaisan et al., 2020). Our experiment shows no significant difference between expert and lay participants in our task — both perform similarly well, and even better on natural images: a replication of our main finding. While a few caveats remain when moving an experiment from the well-controlled lab to a crowdsourcing platform (Haghiri et al., 2019), this suggests that future studies may not have to rely on selected expert participants, but may leverage larger lay participant pools.

(3.) *Are hand-picked synthetic feature visualizations representative?* An open question was whether the visualizations shown in publications represent the general interpretability of feature visualizations (a concern voiced by e.g. Kriegeskorte, 2015), even though they are hand-picked (Olah et al., 2017). Our finding that there is no large difference in performance between hand- and randomly-picked feature visualizations suggests that this aspect is minor.

(4.) *What is the best way of presenting images?* Existing work suggests that more than one example (Offert, 2017) and particularly negative examples (Kim et al., 2016) enhance human understanding of data distributions. Our systematic exploration of presentation schemes provides evidence that increasing the number of reference images as well as presenting both minimally *and* maximally activating reference images (as opposed to only maximally activating ones) improve human performance. This finding might be of interest to future studies aiming at peak performance or for developing software for understanding CNNs.

(5.) *How do humans subjectively perceive feature visualizations?* Apart from the high informativeness of explanations, another relevant question is how much trust humans have in them. In our experiment, we find that subjective impressions of how reasonable synthetic feature visualizations are for explaining responses to natural images vary greatly. This finding is in line with Hase & Bansal (2020) who evaluated explanation methods on text and tabular data.

**Caveats.** Despite our best intentions, a few caveats remain: The forward simulation paradigm is only one specific way to measure the informativeness of explanation methods, but does not allow us to make judgments about their helpfulness in other applications such as comparing different CNNs. Further, we emphasize that all experimental design choices were made with the goal to measure the best possible performance. As a consequence, our finding that synthetic reference images help humans predict a network's strongly activating image may not necessarily be representative of a less optimal experimental set-up with e.g. query images corresponding to less extreme feature map activations. Knobs to further de- or increase participant performance remain (e.g. hyper-parameter choices could be tuned to layers). Finally, while we explored one particular method in depth (Olah et al., 2017); it remains an open question whether the results can be replicated for other feature visualizations methods.

**Future directions.** We see many promising future directions. For one, the current study uses query images from extreme opposite ends of a feature map's activation spectrum. For a more fine-grained measure of informativeness, we will study query images that elicit more similar activations. Additionally, future participants could be provided with even *more* information—such as, for example, where a feature map is located in the network. Furthermore, it has been suggested that the combination of synthetic and natural reference images might provide synergistic information to participants (Olah et al., 2017), which could again be studied in our experimental paradigm. Finally, further studies could explore single neuron-centered feature visualizations, combinations of units as well as different network architectures.

Taken together, our results highlight the need for thorough human quantitative evaluations of feature visualizations and suggest that example natural images provide a surprisingly challenging baseline for understanding CNN activations.

## AUTHOR CONTRIBUTIONS

The initiative of investigating human predictability of CNN activations came from WB. JB, WB, MB and TSAW jointly combined it with the idea of investigating human interpretability of feature visualizations. JB led the project. JB, RSZ and JS jointly designed and implemented the experiments (with advice and feedback from TSAW, RG, MB and WB). The data analysis was performed by JB and RSZ (with advice and feedback from RG, TSAW, MB and WB). JB designed, and JB and JS implemented the pilot study. JB conducted the experiments (with help from JS). RSZ performed the statistical significance tests (with advice from TSAW and feedback from JB and RG). MB helped shape the bigger picture and initiated intuitiveness trials. WB provided day-to-day supervision. JB, RSZ and RG wrote the initial version of the manuscript. All authors contributed to the final version of the manuscript.

## ACKNOWLEDGMENTS

We thank Felix A. Wichmann and Isabel Valera for helpful discussions. We further thank Alexander Böttcher and Stefan Sietzen for support as well as helfpul discussions on technical details. Additionally, we thank Chris Olah for clarifications via `slack.distill.pub`. Moreover, we thank Leon Sixt for valuable feedback on the introduction and related work. From our lab, we thank Matthias Kümmerer, Matthias Tangemann, Evgenia Rusak and Ori Press for helping in piloting our experiments, as well as feedback from Evgenia Rusak, Claudio Michaelis, Dylan Paiton and Matthias Kümmerer. And finally, we thank all our participants for taking part in our experiments.

We thank the International Max Planck Research School for Intelligent Systems (IMPRS-IS) for supporting JB, RZ and RG. We acknowledge support from the German Federal Ministry of Education and Research (BMBF) through the Competence Center for Machine Learning (TUE.AI, FKZ 01IS18039A) and the Bernstein Computational Neuroscience Program Tübingen (FKZ: 01GQ1002), the Cluster of Excellence Machine Learning: New Perspectives for Sciences (EXC2064/1), and the German Research Foundation (DFG; SFB 1233, Robust Vision: Inference Principles and Neural Mechanisms, TP3, project number 276693517).

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

# A APPENDIX

## A.1 DETAILS ON METHODS

### A.1.1 HUMAN EXPERIMENTS

In our two human psychophysical studies, we ask humans to predict a feature map's strongly activating image ("forward simulation task", Doshi-Velez & Kim 2017). Answers to the two-alternative forced choice paradigm are recorded together with the participants' confidence level (1: not confident, 2: somewhat confident, 3: very confident, see Fig. 7). Time per trial is unlimited and we record reaction time. After each trial, feedback is given (see Fig. 7). A progress bar at the bottom of the screen indicates how many trials of a block are already completed. As reference images, either synthetic, natural or no reference images are given. The synthetic images are the feature visualizations from the method of Olah et al. (2017). Trials of different reference images are arranged in blocks. Synthetic and natural reference images are alternated, and, in the case of Experiment I, framed by trials without reference images (see Fig. 8A, B). The order of the reference image types is counter-balanced across subjects.

The main trials in the experiments are complemented by practice, catch and intuitiveness trials. To avoid learning effects, we use different feature maps for each trial type per participant. Specifically, *practice trials* give participants the opportunity to familiarize themselves with the task. In order to monitor the attention of participants, *catch trials* appear randomly throughout blocks of main trials. Here, the query images are a copy of one of the reference images, i.e., there is an obvious correct answer (see Fig. 15). This control mechanism allows us to decide whether trial blocks should be excluded from the analysis due to e.g. fatigue. To obtain the participant's subjective impression of the helpfulness of maximally activating images, the experiments are preceded (and also succeeded in the case of Experiment II) by three *intuitiveness trials* (see Fig. 14). Here, participants judge in a slightly different task design how intuitive they consider the synthetic stimuli for the natural stimuli. For more details on the intuitiveness trials, see below.

At the end of the experiment, all expert participants in Experiment I and all lay (but not expert) participants in Experiment II are asked about their strategy and whether it changed over time. The information gained through the first group allows us to understand the variety of cues used and paves the way to identify interesting directions for follow-up experiments. The information gained through the second group allowed comparisons to experts' impressions reported in Experiment I.

**Experiment I** The first experiment focuses on comparing performance of synthetic images to two baselines: natural reference images and no reference images (see Fig. 8A). Screenshots of trials are shown in Fig. 12. In total, 45 feature maps are tested: 36 of these are uniformly sampled from the feature maps of each of the four branches for each of the nine Inception modules. The other nine feature maps are uniformly hand-picked for interpretability from the Inception modules' pooling branch based on the appendix overview selection provided by Olah et al. (2017) or based on our own choices. In the spirit of a *general* statement about the explainability method, different participants see different natural reference and query images, and each participant sees different natural query images for the same feature maps in different reference conditions. To check the consistency of participants' responses, we repeat six randomly chosen main trials for each of the three tested reference image types at the end of the experiment.

**Experiment II** The second experiment (see Fig. 8B) is about testing expert vs. lay participants as well as comparing different presentation schemes[5] (Max 1, Min+Max 1, Max 9 and Min+Max 9, see Fig. 8E). Screenshots of trials are shown in Fig. 13. In total, 80 feature maps are tested: They are uniformly sampled from every second layer with an Inception module of the network (hence a total of 5 instead of 9 layers), and from all four branches of the Inception modules. Given the focus on four different presentation schemes in this experiment, we repeat the sampling method four times without overlap. In terms of reference image types, only synthetic and natural images are tested. Like in Experiment I, different participants see different natural reference and query images.

---

[5]In pilot experiments, we learned that participants preferred 9 over 4 reference images, hence this "default" choice in Experiment I.

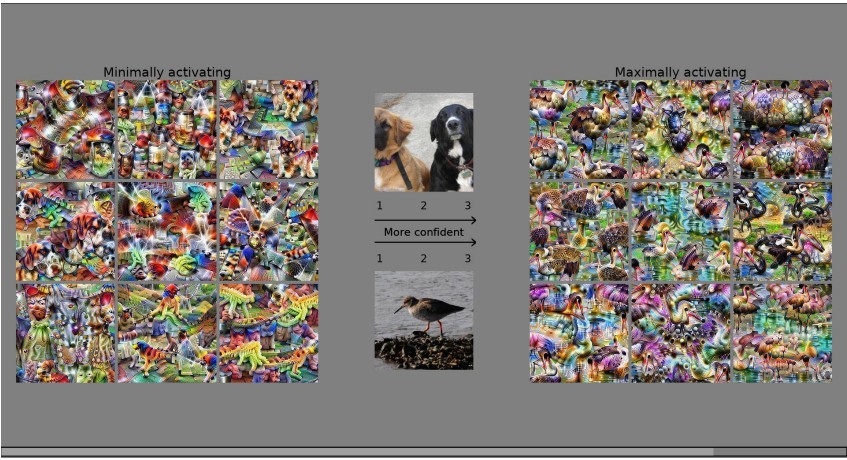

(a) Screen at the beginning of a trial. The question is which of the two natural images at the center of the screen also strongly activates the CNN feature map given the reference images on the sides.

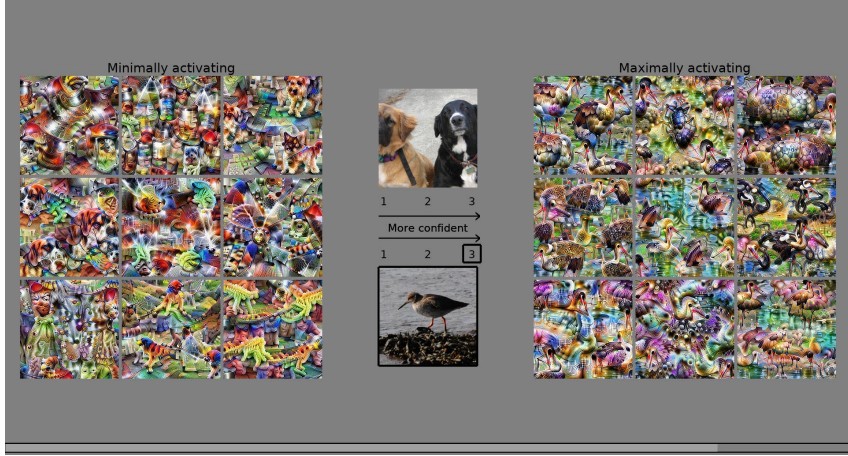

(b) Screen including a participant's answer visualized by black boxes around the image and the confidence level. A participant indicates which natural image at the center would also be a strongly activating image by clicking on the number corresponding to his/her confidence level (1: not confident, 2: somewhat confident, 3: confident). The time until a participant selects an answer is recorded ("reaction time").

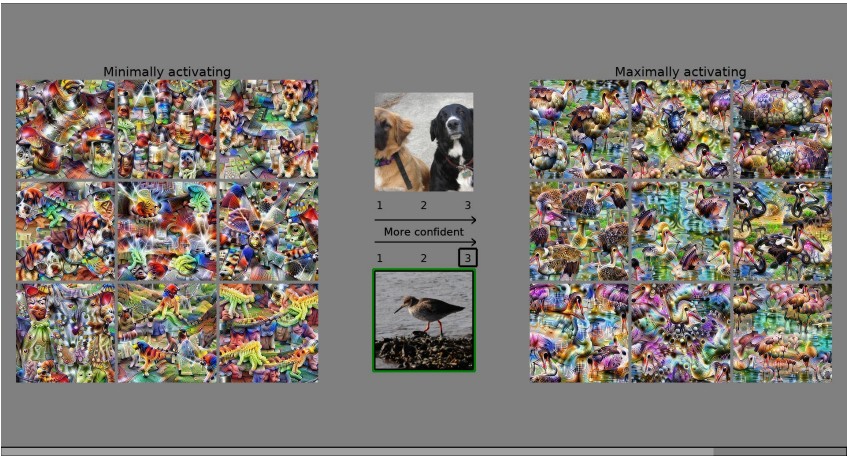

(c) Screen including a participant's answer (black boxes) and feedback on which image is indeed also a strongly activating image (green box).

Figure 7: Forward Simulation Task. The progress bar at the bottom of the screen indicates the progress within one block of trials.

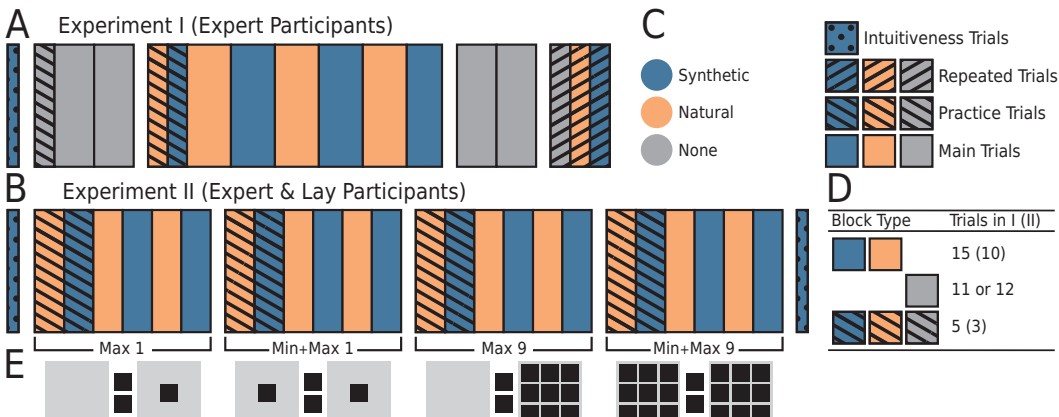

Figure 8: Detailed structure of the two experiments with different foci. **A: Experiment I.** Here, the focus is on comparing performance of synthetic and natural reference images to the most simple baseline: no reference images ("None"). To counter-balance conditions, the order of natural and synthetic blocks is alternated across participants. For each of the three reference image types (synthetic, natural and none), 45 relevant trials are used plus additional catch, practice and repeated trials. **B: Experiment II.** Here, the focus is on testing expert and lay participants as well as comparing different presentation schemes (Max 1, Min+Max 1, Max 9 and Min+Max 9, see **E** for illustrations). Both the order of natural and synthetic blocks as well as the four presentation conditions are counter-balanced across participants. To maintain a reasonable experiment length for each participant, only 20 relevant trials are used per reference image type and presentation scheme, plus additional catch and practice trials. **C:** Legend. **D:** Number of trials per block type (i.e. reference image type and main vs. practice trial) and experiment. Catch trials are not shown in the figure; there was a total of 3 (2) catch trials per each synthetic and natural main block in Experiment I (II). **E:** Illustration of presentation schemes. In Experiment II, all four schemes are tested, in Experiment I only Min+Max 9 is tested.

However, expert and lay participants see the same images. For details on the counter-balancing of all conditions, please refer to Tab. 1.

**Intuitiveness Trials** In order to obtain the participants' subjective impression of the helpfulness of maximally activating images, we add trials at the beginning of the experiments, and also at the end of Experiment II. The task set-up is slightly different (see Fig. 14): Only maximally activating (i.e. no minimally activating) images are shown. We ask participants to rate how intuitive they find the explanation of the entirety of the synthetic images for the entirety of the natural images. Again, all images presented in one trial are specific to one feature map. By moving a slider to the right (left), participants judge the explanation method as intuitive (not intuitive). The ratings are recorded on a continuous scale from $-100$ (not intuitive) to $+100$ (intuitive). All participants see the same three trials in a randomized order. The trials are again taken from the hand-picked (i.e. interpretable) feature maps of the appendix overview in Olah et al. (2017). In theory, this again allows for the highest intuitiveness ratings possible. The specific feature maps are from a low, intermediate and high layer: feature map 43 of mixed3a, feature map 504 of mixed4b and feature map 17 of mixed 5b.

**Participants** Our two experiments are within-subject studies, meaning that every participant answers trials for all conditions. This design choice allows us to test fewer participants. In Experiment I, 10 expert participants take part (7 male, 3 female, age: 27.2 years, SD = 1.75). In Experiment II, 23 participants take part (of which 10 are experts; 14 male, 9 female, age: 28.1 years, SD = 6.76). Expert participants qualify by being familiar or having worked with convolutional neural networks and most of them even with feature visualization techniques. All participants are naive with respect to the aim of the study. Expert (lay) participants are paid 15€ (10 €), per hour for participation. Before the experiment, all participants give written informed consent for participating. All participants have normal or corrected to normal vision. All procedures conform to Standard

8 of the American Psychological 405 Association's "Ethical Principles of Psychologists and Code of Conduct" (2016). Before the experiment, the first author explains the task to each participant and ensures complete understanding. For lay participants, the explanation is simplified: Maximally (minimally) activating images are called "favorite images" ("non-favorite images") of a "computer program" and the question is explained as which of the two query images would also be a "favorite" image to the computer program.

**Apparatus**   Stimuli are displayed on a VIEWPixx 3D LCD (VPIXX Technologies; spatial resolution $1920 \times 1080$ px, temporal resolution $120\,\mathrm{Hz}$). Outside the stimulus image, the monitor is set to mean gray. Participants view the display from $60\,\mathrm{cm}$ (maintained via a chinrest) in a darkened chamber. At this distance, pixels subtend approximately $0.024°$ degrees on average ($41\,\mathrm{ps}$ per degree of visual angle). Stimulus presentation and data collection is controlled via a desktop computer (Intel Core i5-4460 CPU, AMD Radeon R9 380 GPU) running Ubuntu Linux (16.04 LTS), using PsychoPy (Peirce et al., 2019, version 3.0) under Python 3.6.

A.1.2   STIMULI SELECTION

**Model**   Following the existing work on feature visualization by Olah et al. (2017; 2018; 2020b;a), we use an Inception V1 network[6] (Szegedy et al., 2015) trained on ImageNet (Deng et al., 2009; Russakovsky et al., 2015). Note that the Inception V1 network used in previously mentioned work slightly deviates from the original network architecture: The $3 \times 3$ branch of Inception module mixed4a only holds 204 instead of 208 feature maps. To stay as close as possible to the aforementioned work, we also use their implementation and trained weights of the network[7]. We investigate feature visualizations for all branches (i.e. kernel sizes) of the Inception modules and sample from layers mixed3a to mixed5b before the ReLU non-linearity.

**Synthethic Images from Feature Visualization**   The synthetic images throughout this study are the optimization results of the feature visualization method from Olah et al. (2017). We use the channel objective to find synthetic stimuli that maximally (minimally) activate the spatial mean of a given feature map of the network. We perform the optimization using lucid 0.3.8 and TensorFlow 1.15.0 (Abadi et al., 2015) and use the hyperparameter as specified in Olah et al. (2017). For the experimental conditions with more than one minimally/maximally activating reference image, we add a diversity regulariztion across the samples. In hindsight, we realized that we generated 10 synthetic images in Experiment I, even though we only needed and used 9 per feature map.

**Selection of Natural Images**   The natural stimuli are selected from the validation set of the ImageNet ILSVRC 2012 (Russakovsky et al., 2015) dataset. To choose the maximally (minimally) activating natural stimuli for a given feature map, we perform three steps, which are illustrated in Fig. 9 and explained in the following: First, we calculate the activation of said feature map for all pre-processed images (resizing to $256 \times 256$ pixels, cropping centrally to $224 \times 224$ pixels and normalizing) and take the spatial average to get a scalar representing the excitability of the given feature map caused by the image. Second, we order the images according to the collected activation values and select the $(N_{stimuli}+1) \cdot N_{batches}$ maximally (respectively minimally) activating images. Here, $N_{stimuli}$ corresponds to the number of reference images used (either 1 or 9, see Fig. 8, **E**), the $+1$ comes from the query image, and $N_{batches} = 20$ determines the maximum number of participants we can test with our setup. Third, we distribute the selected images into $N_{stimuli}+1$ blocks. Within each block, we randomly shuffle the order of the images. Lastly, we create $N_{batches}$ batches of data by selecting one image from each of the blocks for every batch.[8]

---

[6]This network is considered very interpretable (Olah et al., 2018), yet other work also finds deeper networks more interpretable (Bau et al., 2017). More recent work, again, suggests that "analogous features [...] form across models [...]," i.e. that interpretable feature visualizations appear "universally" for different CNNs (Olah et al., 2020b; OpenAI, 2020).

[7]`github.com/tensorflow/lucid/tree/v0.3.8/lucid`

[8]After having performed Experiment I and II, we realized a minor bug in our code: Instead of moving every 20[th] image into the same batch for one participant, we moved every 10[th] image into the same batch for one participant. This means that we only use a total of 110 different images, instead of 200. The minimal query image is still always selected from the 20 least activating images; the maximal query image is selected from the 91[st] to 110[th] maximally activating images - and we do not use the 111[th] to 200[th] maximally activating images.

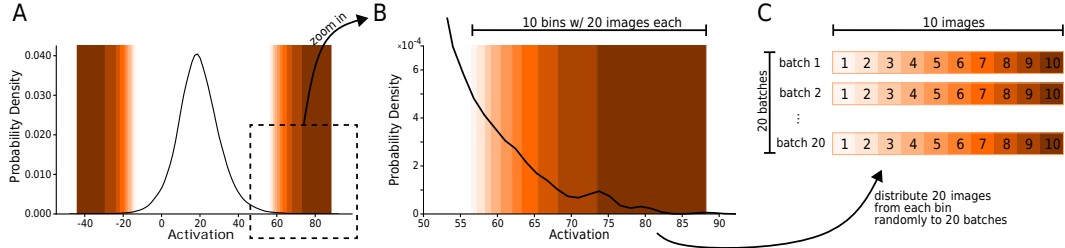

Figure 9: Sampling of natural images. **A:** Distribution of activations. For an example channel (mixed3a, kernel size $1 \times 1$, feature map 25), the smoothed distribution of activations for all $50,000$ ImageNet validation images is plotted. The natural stimuli for the experiment are taken from the tails of the distribution (shaded background). **B:** Zoomed-in tail of activations distribution. In the presentation schemes with 9 images, 10 bins with 20 images each are created (10 because of 9 reference plus 1 query image). **C:** In order to obtain 20 batches with 10 images each, the 20 images from one bin are randomly distributed to the 20 batches. This guarantees that each batch contains a fair selection of extremely activating images. The query images are *always* sampled from the most extreme bins in order to give the best signal possible. In the case of the presentation schemes with 1 reference image, the number of bins in B is reduced to 2 and the number of images per batch in C is also reduced to 2.

| Subject | Order of presentation schemes (0-3) and batch-blocks (A-D) | | | | Batches Practice | Main | Order of synthetic and natural |
|---|---|---|---|---|---|---|---|
| 1 | 0 (A) | 1 (B) | 2 (C) | 3 (D) | | | |
| 2 | 0 (B) | 2 (D) | 1 (C) | 3 (A) | | natural: 1 | natural - synthetic |
| 3 | 3 (B) | 1 (D) | 2 (A) | 0 (C) | | synthetic: 2 | |
| 4 | 3 (C) | 2 (B) | 1 (A) | 0 (D) | | | |
| 5 | | | | | | | |
| 6 | | see subject 1-4 | | | | natural: 3 | synthetic - natural |
| 7 | | | | | 0 | synthetic: 4 | |
| 8 | | | | | | | |
| 9 | | | | | | | |
| 10 | | see subject 1-4 | | | | natural: 5 | natural - synthetic |
| 11 | | | | | | synthetic: 6 | |
| 12 | | | | | | | |
| 13 | | see subject 1-4 | | | | natural: 7 synthetic: 8 | synthetic - natural |

Table 1: **Counter-balancing of conditions in Experiment II.** In total, 13 naive and 10 lay participants are tested. Each "batch block" contains 20 feature maps (sampled from five layers and all Inception module branches). Batches indicate which batch number the natural query (and reference images) are taken from.

The reasons for creating several batches of extremely activating natural images are two-fold: (1) We want to get a *general* impression of the interpretability method and would like to reduce the dependence on single images, and (2) in Experiment I, a participant has to see different query images in the three different reference conditions. A downside of this design choice is an increase in variability. The precise allocation was done as follows: In Experiment I, the natural query images of the none condition were always allocated the batch with $batch\_nr = subject\_id$, the query and reference images of the natural condition were allocated the batch with $batch\_nr = subject\_id+1$, and the natural query images of the synthetic condition were allocated the batch with $batch\_nr = subject\_id+2$. The allocation scheme in Experiment II can be found in Table 1.

**Selection of Feature Maps** The selection of feature maps used in Experiment I is shown in Table 2; the selection of feature maps used in Experiment II is shown in Table 3.

| Layer | Branch | Feature Map | Layer | Branch | Feature Map |
|---|---|---|---|---|---|
| mixed3a | $1 \times 1$ | 25 | mixed4d | $1 \times 1$ | 95 |
| | $3 \times 3$ | 189 | | $3 \times 3$ | 342 |
| | $5 \times 5$ | 197 | | $5 \times 5$ | 451 |
| | Pool | 227 | | Pool | 483 |
| | Pool* | 230 | | Pool* | 516 |
| mixed3b | $1 \times 1$ | 64 | mixed4e | $1 \times 1$ | 231 |
| | $3 \times 3$ | 178 | | $3 \times 3$ | 524 |
| | $5 \times 5$ | 390 | | $5 \times 5$ | 656 |
| | Pool | 430 | | Pool | 816 |
| | Pool* | 462 | | Pool* | 809 |
| mixed4a | $1 \times 1$ | 68 | mixed5a | $1 \times 1$ | 229 |
| | $3 \times 3$ | 257 | | $3 \times 3$ | 278 |
| | $5 \times 5$ | 427 | | $5 \times 5$ | 636 |
| | Pool | 486 | | Pool | 743 |
| | Pool* | 501 | | Pool* | 720 |
| mixed4b | $1 \times 1$ | 45 | mixed5b | $1 \times 1$ | 119 |
| | $3 \times 3$ | 339 | | $3 \times 3$ | 684 |
| | $5 \times 5$ | 438 | | $5 \times 5$ | 844 |
| | Pool | 491 | | Pool | 1007 |
| | Pool* | 465 | | Pool* | 946 |
| mixed4c | $1 \times 1$ | 94 | | | |
| | $3 \times 3$ | 247 | | | |
| | $5 \times 5$ | 432 | | | |
| | Pool | 496 | | | |
| | Pool* | 449 | | | |

Table 2: Feature maps analyzed in Experiment I. For each of the 9 layers with an Inception module, one randomly chosen feature map per branch ($1 \times 1$, $3 \times 3$, $5 \times 5$ and pool) and one additional hand-picked feature map (highlighted with *) are used.

### A.1.3 DIFFERENT ACTIVATION MAGNITUDES

We note that the elicited activations of synthetic images are almost always about one magnitude larger than the activations of natural images (see Fig. 10a). This constitutes an inherent difference in the synthetic and natural reference image condition. A simple approach to make the two conditions more comparable is to limit the optimization process such that the resulting feature visualizations elicit activations similar to that of natural images. This can be achieved by halting the optimization process once the activations approximately match. By following that procedure one finds limited synthetic images which are indistinguishable from natural images in terms of their activations (see Fig. 10b). Importantly though, these images are visually not more similar to natural images, have a much lower color contrast than normal feature visualizations, and above all hardly resemble meaningful features (see Fig. 11).

### A.1.4 DATA ANALYSIS

**Significance Tests** All significance tests are performed with JASP (JASP Team, 2020, version 0.13.1). For the analysis of the distribution of confidence ratings (see Fig. 3B), we use contingency tables with $\chi^2$-tests. For testing pairwise effects in accuracy, confidence, reaction time and intuitiveness data, we report Wilcoxon signed-rank tests with uncorrected p-values (Bonferroni-corrected critical alpha values with family-wise alpha level of $0.05$ reported in all figures where relevant). These non-parametric tests are preferred for these data because they do not make distributional assumptions like normally-distributed errors, as in e.g. paired $t$-tests. For testing marginal effects (main effects of one factor marginalizing over another) we report results from repeated measures ANOVA (RM ANOVA), which does assume normality.

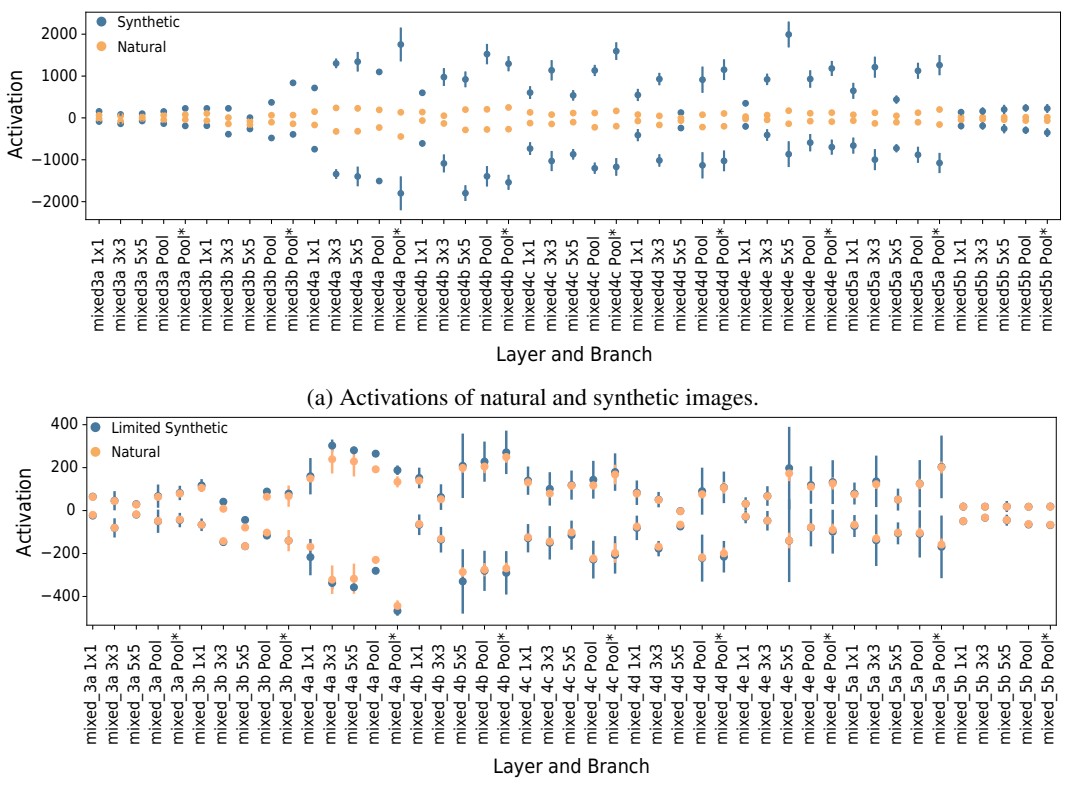

(a) Activations of natural and synthetic images.

(b) Activations of natural and limited synthetic images.

Figure 10: Mean activations and standard deviations (not two standard errors of the mean!) of the minimally (below 0) and maximally (above 0) activating synthetic and natural images used in Experiment I. Note that there are 10 (i.e. accidentally not 9) synthetic images and $20 \cdot 10 = 200$ natural images (because of 20 batches) in Experiment I for both minimally and maximally activating images. Please also note that the standard deviations for the selected natural images are invisible because they are so small. Limited synthetic images refer to feature visualizations which are the result of stopping the optimization process early with the goal of matching the activation level of natural stimuli.

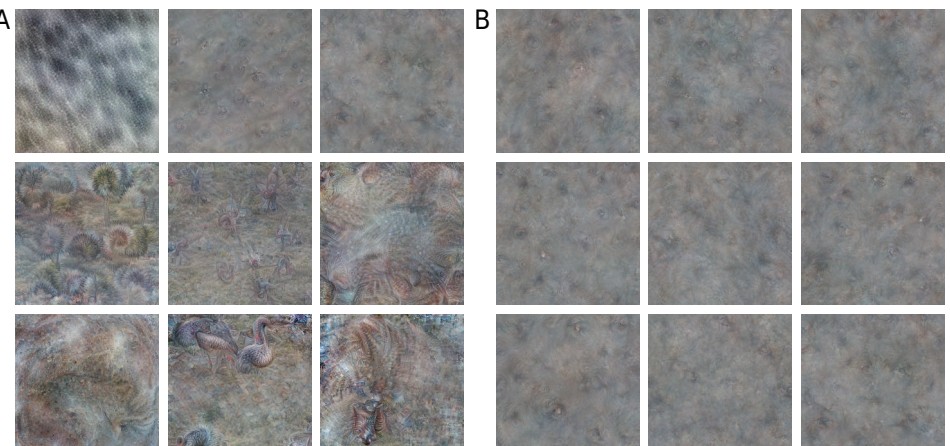

Figure 11: Limited feature visualizations, which are the result of stopping the optimization process early with the goal of matching the activation level of the chosen extreme natural stimuli. **A**: Feature visualizations for mixed_4a pool* feature map of Experiment I. **B**: Feature visualizations for all nine pool* feature maps of Experiment I.

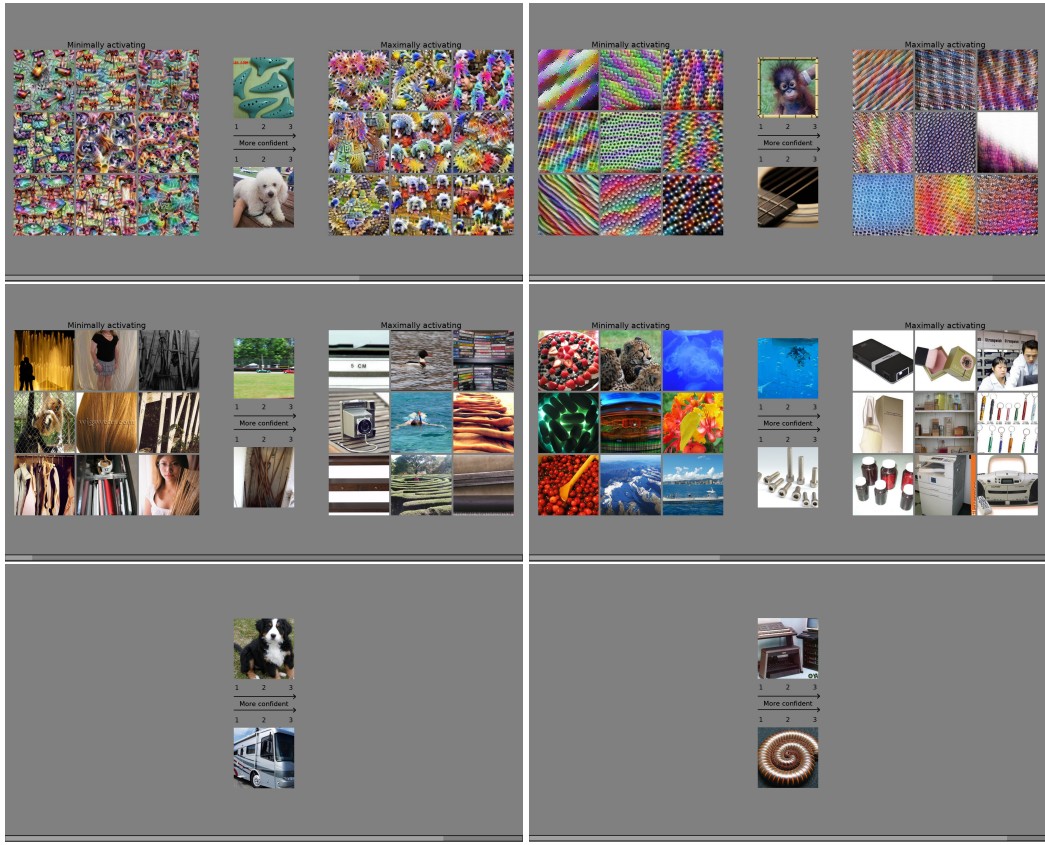

Figure 12: Experiment I: Example trials of the three reference images conditions: synthetic reference images (first row), natural reference images (second row) or no reference images (third row). The query images in the center are always natural images.

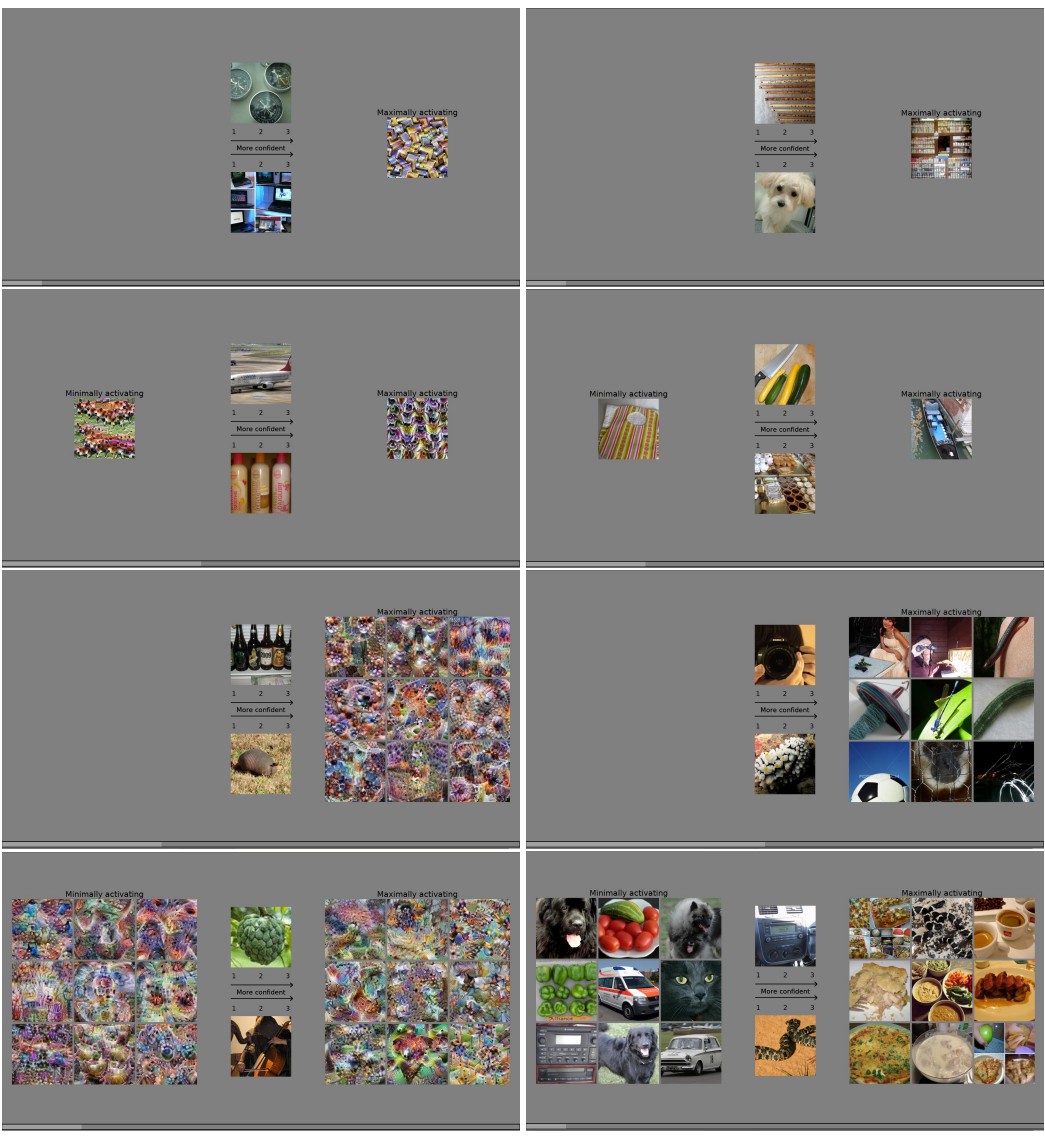

Figure 13: Experiment II: Example trials of the four presentation schemes: Max 1, Min+max 1, Max 9, Min+Max 9. The left column contains synthetic reference images, the right column contains natural reference images.

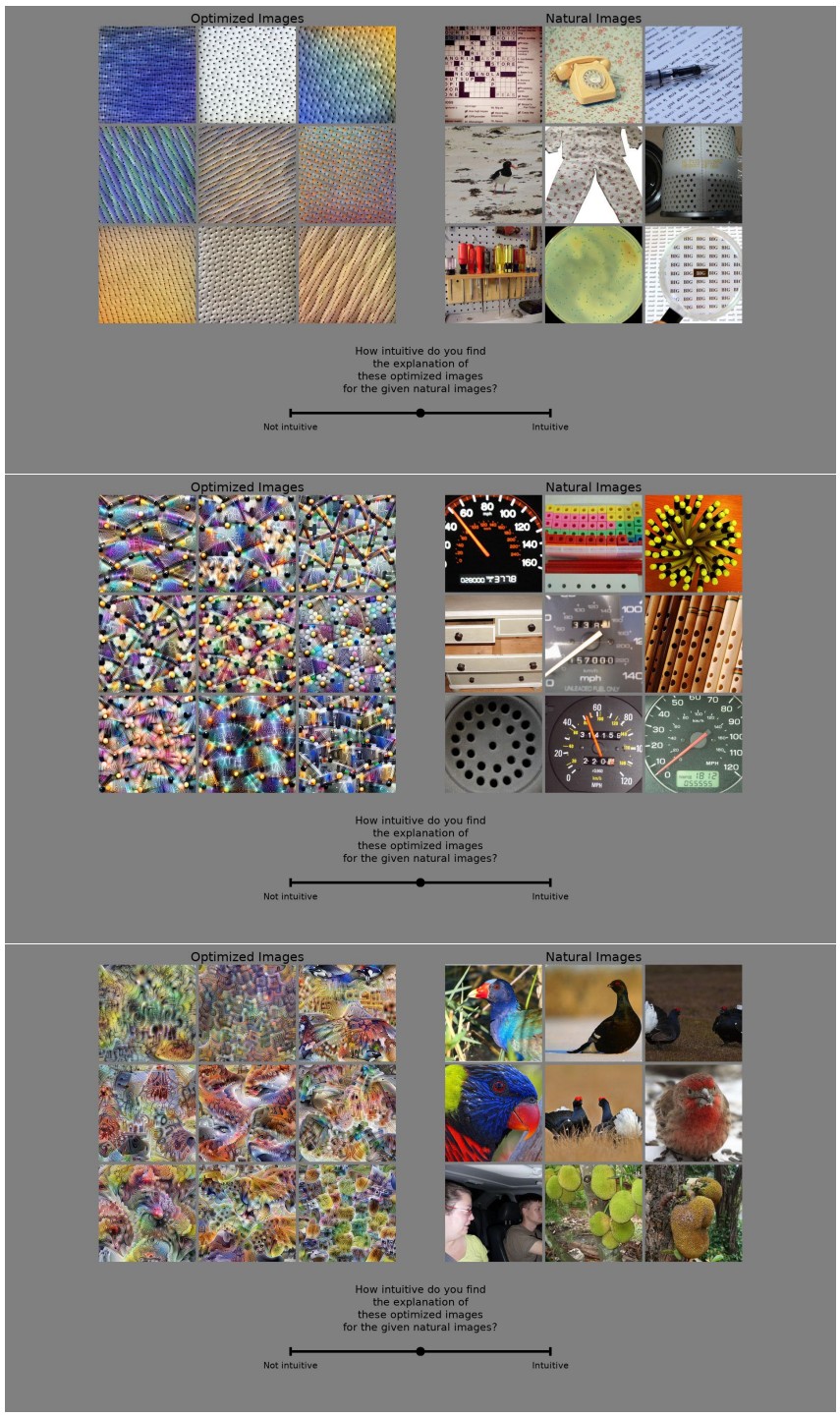

Figure 14: Trials for intuitiveness judgment. The tested feature maps are from layer mixed3a (channel 43), mixed4b (channel 504) and mixed 5b (channel 17). They are the same in Experiment I and in Experiment II.

| Layer | Branch | Feature Map for Batch Block (A-D) | | | |
|---|---|---|---|---|---|
| | | A | B | C | D |
| mixed3a | $1 \times 1$ | 25 | 14 | 12 | 53 |
| | $3 \times 3$ | 189 | 97 | 171 | 106 |
| | $5 \times 5$ | 197 | 203 | 212 | 204 |
| | Pool | 227 | 238 | 232 | 247 |
| mixed4a | $1 \times 1$ | 68 | 33 | 45 | 17 |
| | $3 \times 3$ | 257 | 355 | 321 | 200 |
| | $5 \times 5$ | 427 | 425 | 429 | 423 |
| | Pool | 486 | 497 | 478 | 506 |
| mixed4c | $1 \times 1$ | 94 | 53 | 59 | 95 |
| | $3 \times 3$ | 247 | 237 | 357 | 209 |
| | $5 \times 5$ | 432 | 402 | 400 | 416 |
| | Pool | 496 | 498 | 473 | 497 |
| mixed4e | $1 \times 1$ | 231 | 83 | 6 | 89 |
| | $3 \times 3$ | 524 | 323 | 401 | 373 |
| | $5 \times 5$ | 656 | 624 | 642 | 620 |
| | Pool | 816 | 755 | 724 | 783 |
| mixed5b | $1 \times 1$ | 119 | 14 | 266 | 300 |
| | $3 \times 3$ | 684 | 592 | 657 | 481 |
| | $5 \times 5$ | 844 | 829 | 839 | 875 |
| | Pool | 1007 | 913 | 927 | 903 |

Table 3: Feature maps analyzed in Experiment II. Four sets of feature maps (batch blocks A to D) are sampled: For every second layer with an Inception module (5 layers in total), one feature map is randomly selected per branch of the Inception module ($1 \times 1$, $3 \times 3$, $5 \times 5$ and pool). For the practice, catch and intuitiveness trials additional randomly chosen feature maps are used.

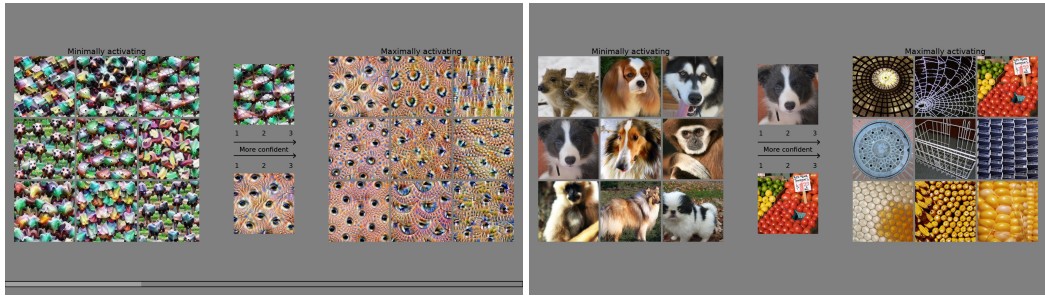

Figure 15: Catch trials. An image from the reference images is copied as a query image, which makes the answer obvious. The purpose of these trials is to integrate a mechanism into the experiment which allows us to check post-hoc whether a participant was still paying attention.

## A.2 DETAILS ON RESULTS

### A.2.1 COMPLEMENTING FIGURES FOR MAIN RESULTS

Figures 16 - 21 complement the results and figures presented in Section 4. Here, all experimental conditions are shown.

### A.2.2 DETAILS ON PERFORMANCE OF EXPERT AND LAY PARTICIPANTS

As reported in the main body of the paper, a mixed-effects ANOVA revealed no significant main effect of expert level ($F(1, 21) = 0.6$, $p = 0.44$, between-subjects effect). Further, there is no significant interaction with the reference image type ($F(1, 21) = 0.4, p = 0.53$), and both expert and lay participants show a significant main effect of the reference image type ($F(1, 21) = 230.2, p < 0.001$).

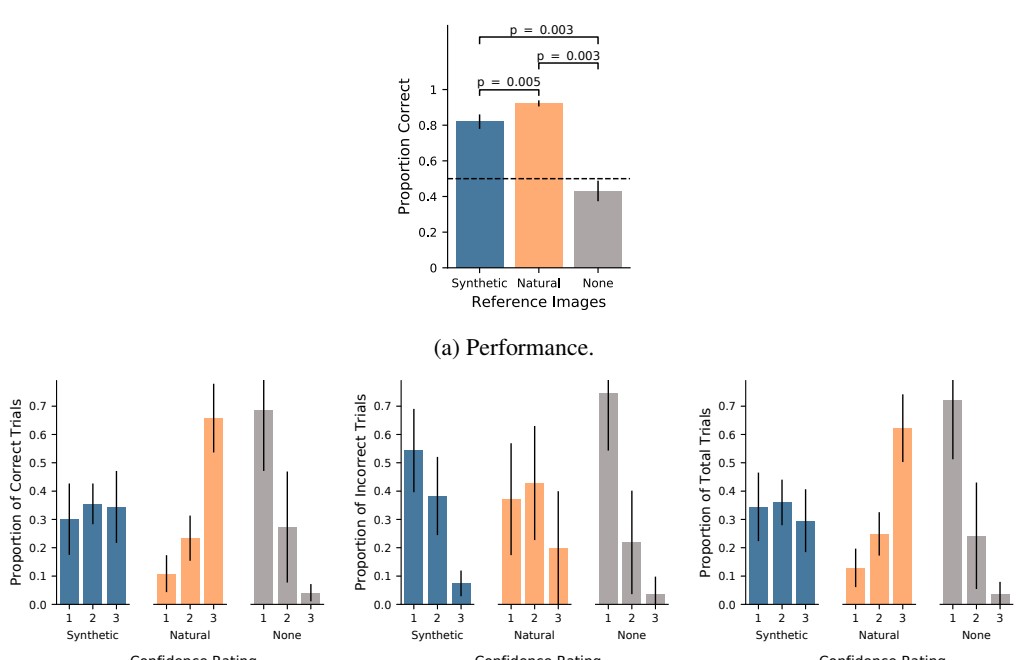

(a) Performance.

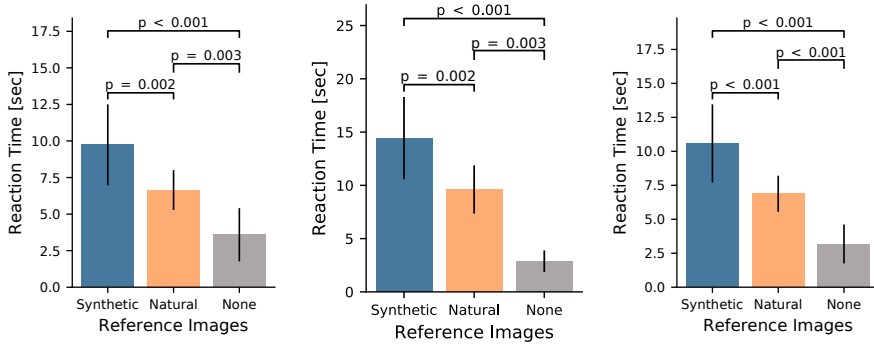

(b) Confidence ratings on correctly answered trials.

(c) Confidence ratings on incorrectly answered trials.

(d) Confidence ratings on all trials.

(e) Reaction time on correctly answered trials.

(f) Reaction time on incorrectly answered trials.

(g) Reaction time on all trials.

Figure 16: Task performance (a), distribution of confidence ratings (b-d) and reaction times (e-g) of Experiment I. The $p$-values are calculated with Wilcoxon sign-rank tests. Note that unlike in the main paper, these figures consistently include the "None" condition. For explanations, see Sec. 4.1.

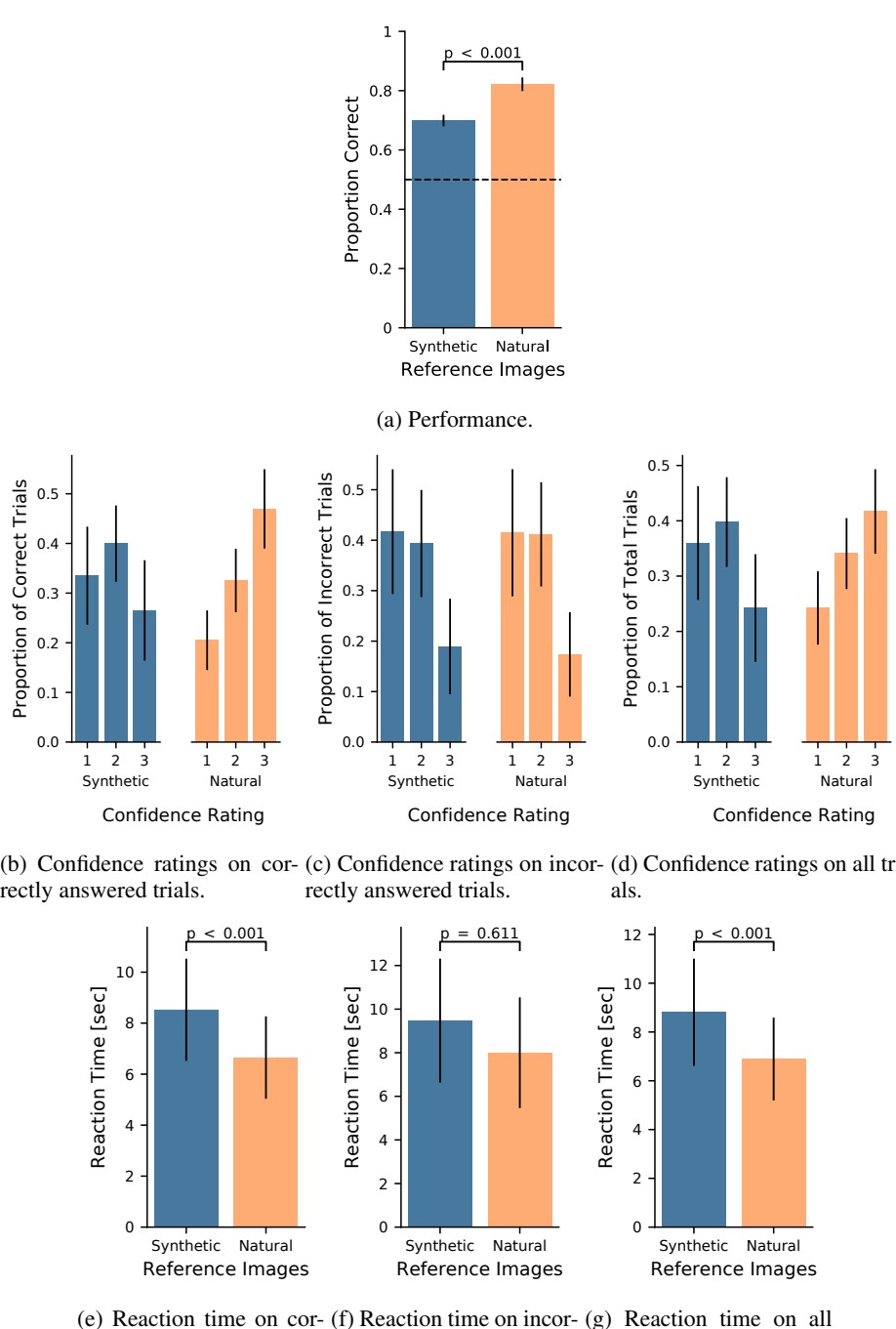

(a) Performance.

(b) Confidence ratings on correctly answered trials.
(c) Confidence ratings on incorrectly answered trials.
(d) Confidence ratings on all trials.

(e) Reaction time on correctly answered trials.
(f) Reaction time on incorrectly answered trials.
(g) Reaction time on all trials.

Figure 17: Task performance (a), distribution of confidence ratings (b-d) and reaction times (e-g) of Experiment II, averaged over expert level and presentation schemes. The $p$-values are calculated with Wilcoxon sign-rank tests. The results replicate our findings of Experiment I. For explanations on the latter, see Sec. 4.1.

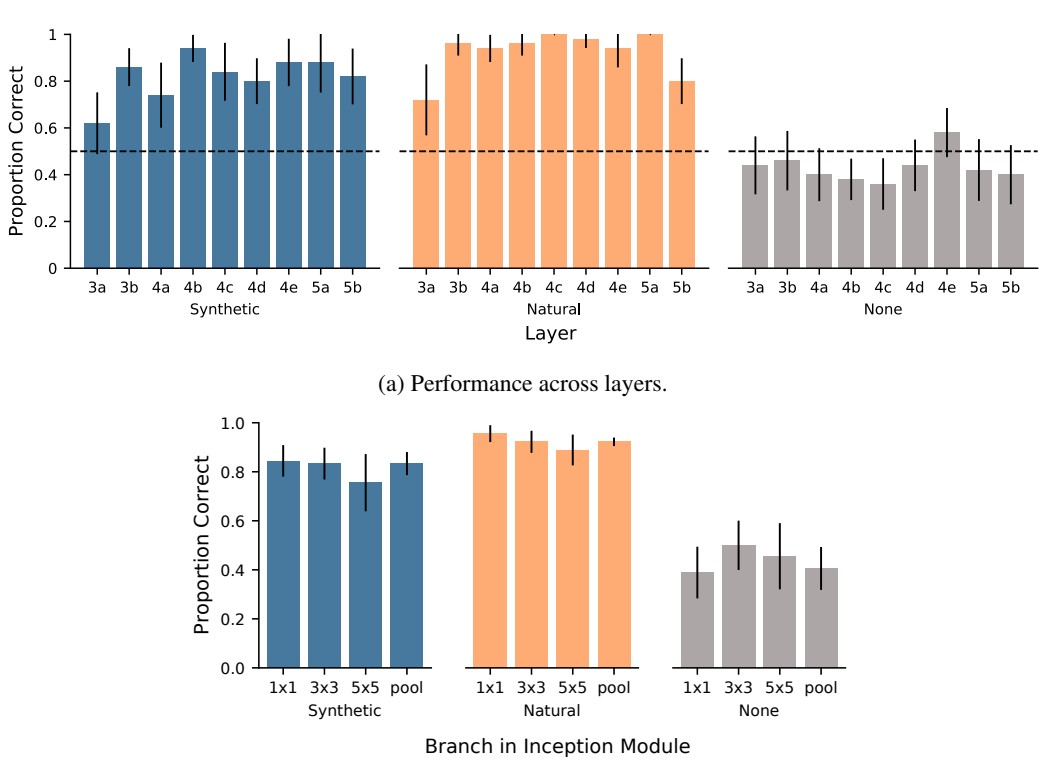

(a) Performance across layers.

(b) Performance across branches.

Figure 18: High performance across (a) layers and (b) branches of the Inception modules in Experiment I. Note that unlike in the main paper these figures consistently include the "None" condition. For explanations, see Sec. 4.2.

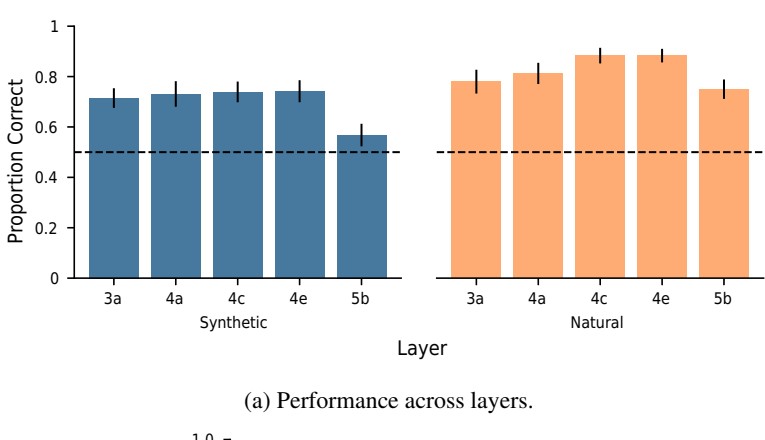

(a) Performance across layers.

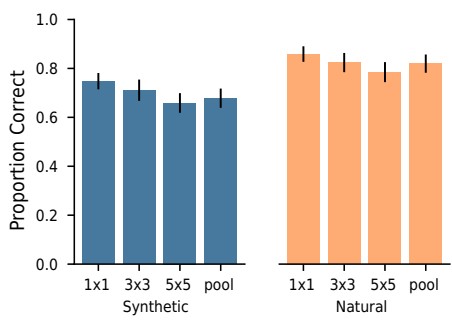

(b) Performance across branches in Inception module.

Figure 19: High performance across (a) layers and (b) branches of the Inception modules in Experiment II. Note that only every second layer is tested here (unlike in Experiment I). The results replicate our findings of Experiment I. For explanations, see Sec. 4.2

### A.2.3 Details on performance of experts split by different levels of expertise

Even though Experiment II does not show a significant performance difference for lay and expert participants, it is an open question whether the level of expertise or the background of experts matters. For the data from experts, we hence further divide participants into subgroups according to their expertise (see Fig. 20a-f) and background level (see Fig. 20g-h). Expertise level 1 means that participants are familiar with CNNs, but not feature visualizations; expertise level 2 means that participants have heard of or read about feature visualizations; and expertise level 3 means that participants have used feature visualizations themselves. We note that we also accepted feature visualizations methods other than the one by Olah et al. (2017), e.g. DeepDream (Mordvintsev et al., 2015) for level 2 and 3. Regarding background, we distinguished computational neuroscientists from researchers working on computer vision and / or machine learning. We note that some subgroups only hold one participant and hence may not be representative.

Our data shows varying trends for the three expert levels (see Fig. 20a-f): For synthetic images, performance decreases with increasing expertise in Experiment I, but increases for Experiment II. For natural images, performance first increases for participants of expertise level 2, and then slightly decreases for participants with expertise level 3 - a trend that holds for both Experiment I and II. In the none condition of Experiment I, performance is highest for the participant of expertise level 1, but decreases for participants of expertise level 2, and again slightly increases for expertise level 3.

Regarding expert's different backgrounds, our hypothesis is that many of the computational neuroscientists are very familiar with maximally exciting images for monkeys or rodents, and hence might perform better than pure computer vision / machine learning experts. Fig. 20g-h suggest that this is not the case: The bars for all three reference image types are very similar.

Not finding clear trends in our data between different expertise levels or experts is not surprising as there is even no significant difference between participants whose professional backgrounds are much further apart: lay people vs. people familiar with CNNs.

### A.2.4 Details on performance of hand- and randomly-picked feature maps

As described in the main body of the paper, pairwise Wilcoxon sign-rank tests reveal no significant differences between hand-picked and randomly-selected feature maps within each reference image type ($Z(9) = 27.5$, $p = 0.59$ for natural reference images and $Z(9) = 41$ $p = 0.18$ for synthetic references). However, marginalizing over reference image type using a repeated measures ANOVA reveals a significant main effect of the feature map selection mode: $F(1, 9) = 6.14, p = 0.035$. Therefore, while there may be a small effect of hand-picking feature maps, our data indicates that this effect, if present, is small.

### A.2.5 Repeated trials

To check the consistency of participants' responses, we repeat six main trials for each of the three tested reference image types at the end of the experiment. Specifically, the six trials correspond to the three highest and three lowest absolute confidence ratings. Results are shown in Fig. 21. We observe consistency to be high for both the synthetic and natural reference image types, and moderate for no reference images (see Fig. 21A). In absolute terms, the largest increase in performance occurs for the none condition; for natural reference images there was also a small increase; for synthetic reference images, there was a slight decrease (see Fig. 21B and C). In the question session after the experiments, many participants reported remembering the repeated trials from the first time.

### A.2.6 Qualitative Findings

In a qualitative interview conducted after completion of the experiment, participants reported to use a large variety of strategies. Colors, edges, repeated patterns, orientations, small local structures and (small) objects were commonly mentioned. Most but not all participants reported to have adapted their decision strategy throughout the experiment. Especially lay participants from Experiment II emphasized that the trial-by-trial feedback was helpful and that it helped to learn new strategies. As already described in the main text, participants reported that the task difficulty varied greatly; while some trials were simple, others were challenging. A few participants highlighted that the comparison between minimally and maximally activating images was a crucial clue and allowed employing the

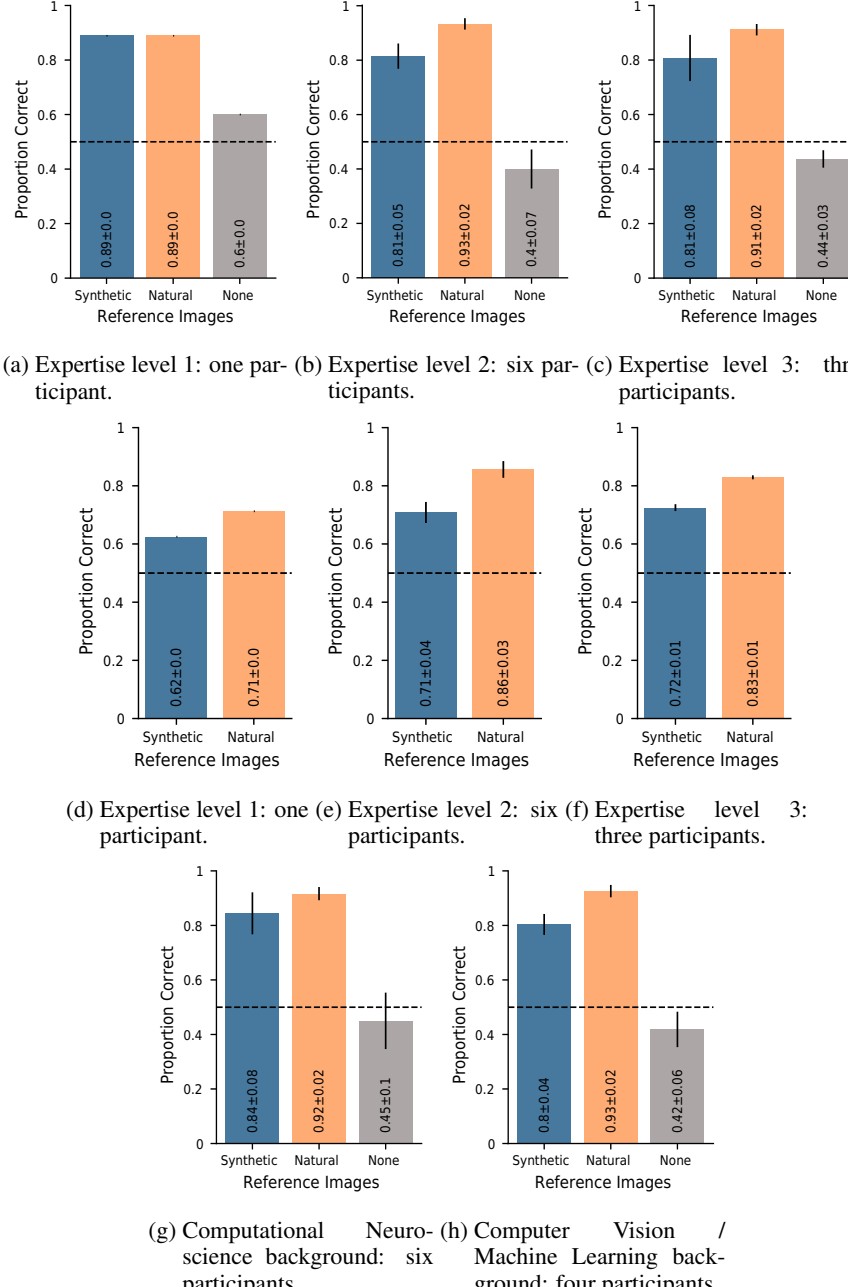

(a) Expertise level 1: one participant.

(b) Expertise level 2: six participants.

(c) Expertise level 3: three participants.

(d) Expertise level 1: one participant.

(e) Expertise level 2: six participants.

(f) Expertise level 3: three participants.

(g) Computational Neuroscience background: six participants.

(h) Computer Vision / Machine Learning background: four participants.

Figure 20: Performance of experts split by different levels of expertise: The first (second) row shows the data of Experiment I (II) split up by different levels of familiarity with CNNs and feature visualizations. The third row shows the data of Experiment I split up by different backgrounds.

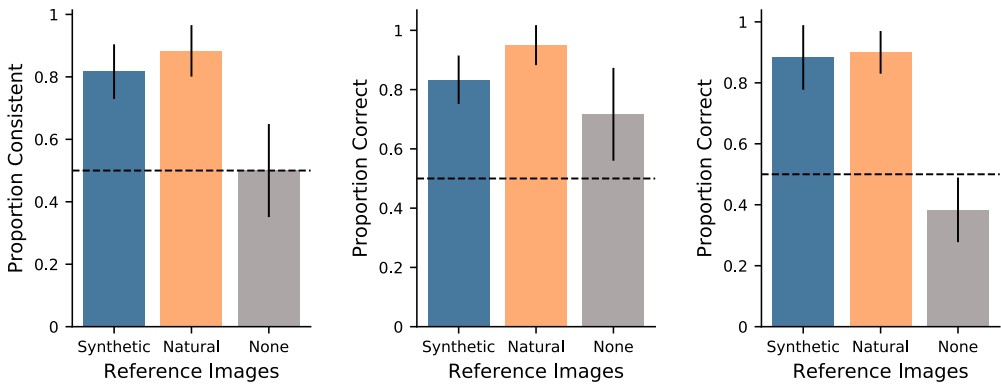

(a) Proportion of trials that were answered the same upon repetition.

(b) Performance for repeated trials upon repetition.

(c) Performance for repeated trials when first shown.

Figure 21: Repeated trials in Experiment I.

exclusion criterion: If the minimally activating query image was easily identifiable, the choice of the maximally activating query image was trivial. This aspect motivated us to conduct an additional experiment where the presentation scheme was varied (Experiment II).

### A.2.7 BY-FEATURE-MAP ANALYSIS

For Experiment I, we look at each feature map separately and analyze which feature maps participants find easy and which they find difficult. Further, we investigate commonalities and differences between feature maps. We note that the data for this analysis relies on only 10 responses for each feature map and hence may be noisy.

In Fig. 22, we show the number of correct answers split up by reference image type. The patterns look similar to the trend in Fig. 4: Across most layers, there is no clearly identifiable trend that feature maps of a certain network depth would be easier or more difficult; only the lowest (3a) and the highest layer (5b) seem slightly more difficult for both the synthetic and the natural reference images.

**Easy Feature Maps**   When feature maps are easy (synthetic: $10/10$, natural: $10/10$ correct responses), their features seem to correspond to clear object parts (e.g. dogs vs. humans, food vs. cats), or shapes (e.g. round vs. edgy (see Supplementary Material Fig. 2- 5)). In Fig. 23, we show the query as well as natural and synthetic reference images for one such easy feature map for one participant. For the images shown to two more participants, see Supplementary Material Fig. 1. Other relatively easy feature maps (where eight to ten participants choose the correct query image for both reference image types) additionally contained other low level cues such as color or texture (see Supplementary Material Fig. 4-5).

**Difficult Feature Maps**   The most difficult feature maps for synthetic and natural reference images are displayed in Fig. 24. Only four participants predicted the correct query image. Interestingly, the other reference image type was much more easily predictable for both feature maps: Nine out of ten participants correctly simulated the network's decision. Our impression is that the reason for these feature maps being so difficult in one reference condition is the diversity in the images. In the case of synthetic reference images, we also consider identifying a concept difficult and consequently are unsure what to compare.

From studying several feature maps, our impression is that one or more of the following aspects make feature maps difficult to interpret:

- Reference images are diverse (see Fig. 24a for synthetic reference images and d for natural reference images)

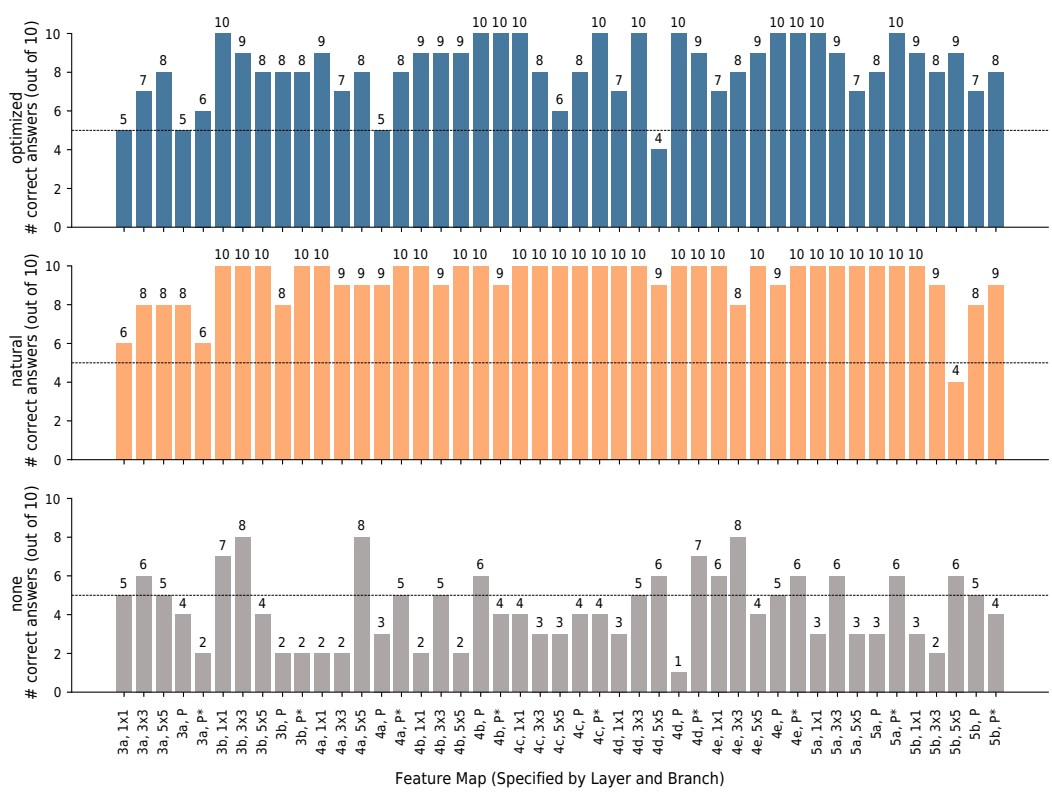

Figure 22: Data for Experiment I split up by feature maps: For each reference image type, the number of correct answers (out of ten) is shown. There is no clear trend that certain feature maps would be easier or more difficult.

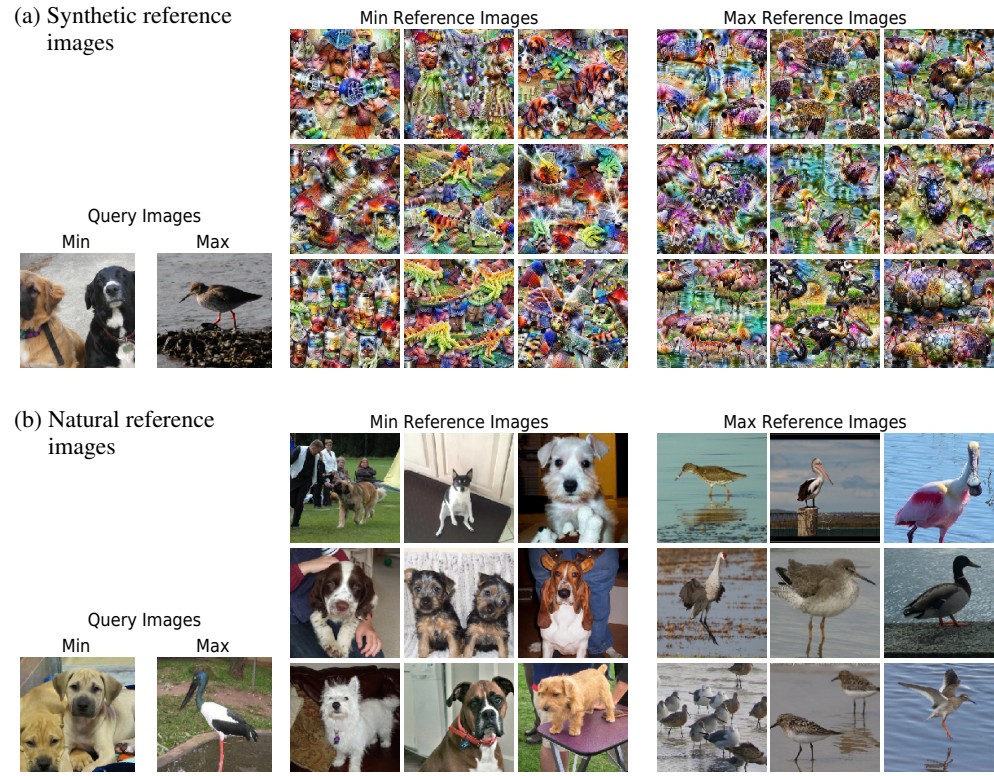

Figure 23: An easy feature map (here: 5a, pool*) from Experiment I where all participants answered correctly for both synthetic and natural reference images. The shown stimuli were shown to participant 1, for stimuli shown to participant 2 and 3, see Supplementary Material Fig 1.

- The common feature(s) seem to not correspond to common human concepts (see Fig. 24a and c)

- Conflicting information, i.e. commonalities can be found between one query image and both the minimal and maximal reference images (see Fig. 25a: eyes and extremity-like structure in synthetic min reference images vs. eyes and earth-colors in synthetic max reference images - both could be considered similar to the max query image of a frog)

- Very small object parts such as eyes or round, earth-colored shapes seem to be the decisive features (see Fig. 25a and b)

- Low level cues such as the orientation of lines appear random in the synthetic reference images[9] (see Fig. 26a)

Finally, when we speak bluntly, we are often surprised that participants identified the correct image — the reasons for this are unclear to us (see for example Supplementary Material Fig. 6-7).

### A.2.8 HIGH QUALITY DATA AS SHOWN BY HIGH PERFORMANCE ON CATCH TRIALS

We integrate a mechanism to probe the quality of our data: In *catch trials*, the correct answer is trivial and hence incorrect answers might suggest the exclusion of specific trial blocks (for details, see Sec. A.1.1). Fortunately, very few trials are missed: In Experiment I, only two (out of ten) participants miss one trial each (i.e. a total of 2 out of 180 catch trials were missed); in Experiment II, five participants miss one trial and four participants miss two trials (i.e. a total of 13 out of 736 catch

---

[9]We expected lower layers to be easier than higher layers for synthetic reference images, but our data showed that this was not the case (see Fig. 22. We can imagine that the diversity term as well as the non-custom hyper-parameters contribute to these sub-optimal images.

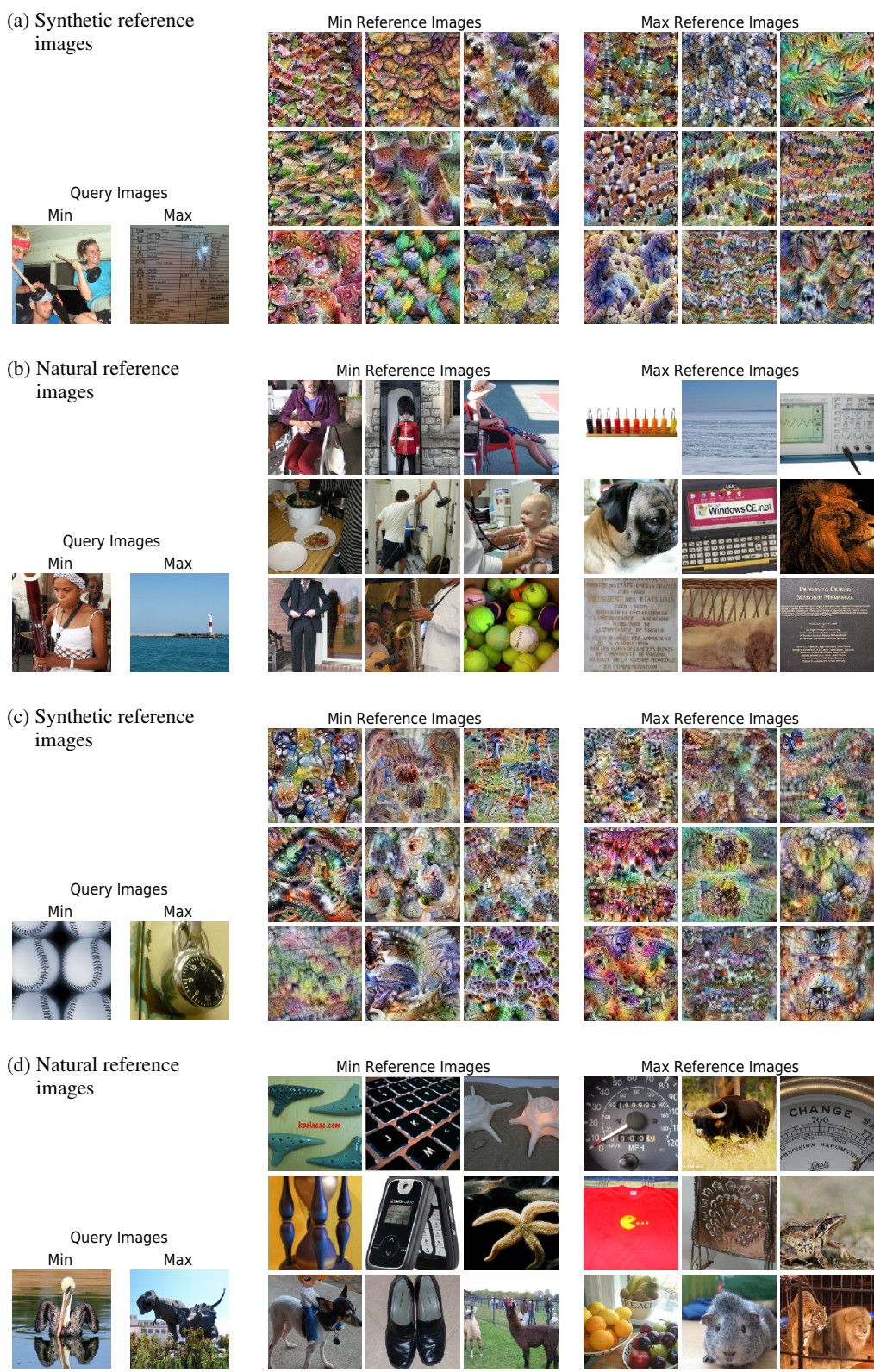

Figure 24: Two difficult feature maps (4d, 5x5 in a and b; 5b, 5x5 in c and d) from Experiment I where only four participants answered correctly for synthetic (a and b) and natural (c and d) reference images. The displayed stimuli were shown to participant 1, for stimuli shown to participant 2 (3), see Supplementary Material Fig. 8 (9).

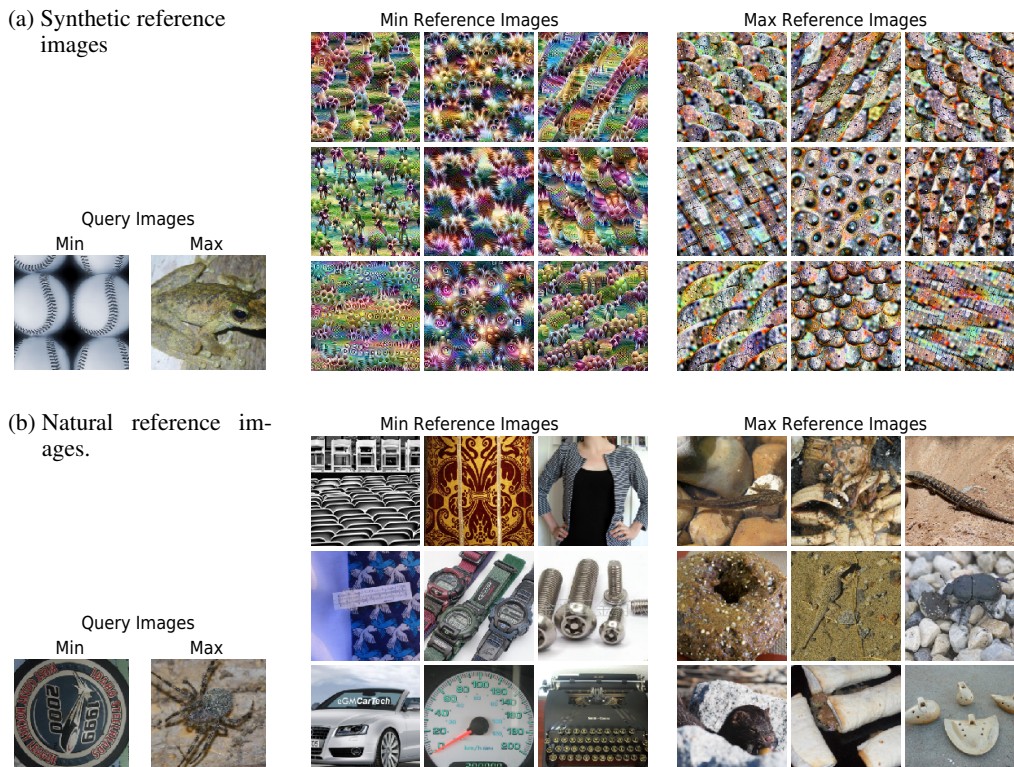

Figure 25: A feature map (here: 4a, Pool) from Experiment I where the feature is small (eyes) and a participant might perceive conflicting information (eyes and extremity-like structure in min reference images vs. eyes and earth-colors in max reference images). In this specific example, eight (nine) out of ten participants gave the correct answer for this feature map given synthetic (natural) reference images. The displayed stimuli were shown to participant 1, for stimuli shown to participant 2 and 3, see Supplementary Material Fig. 10.

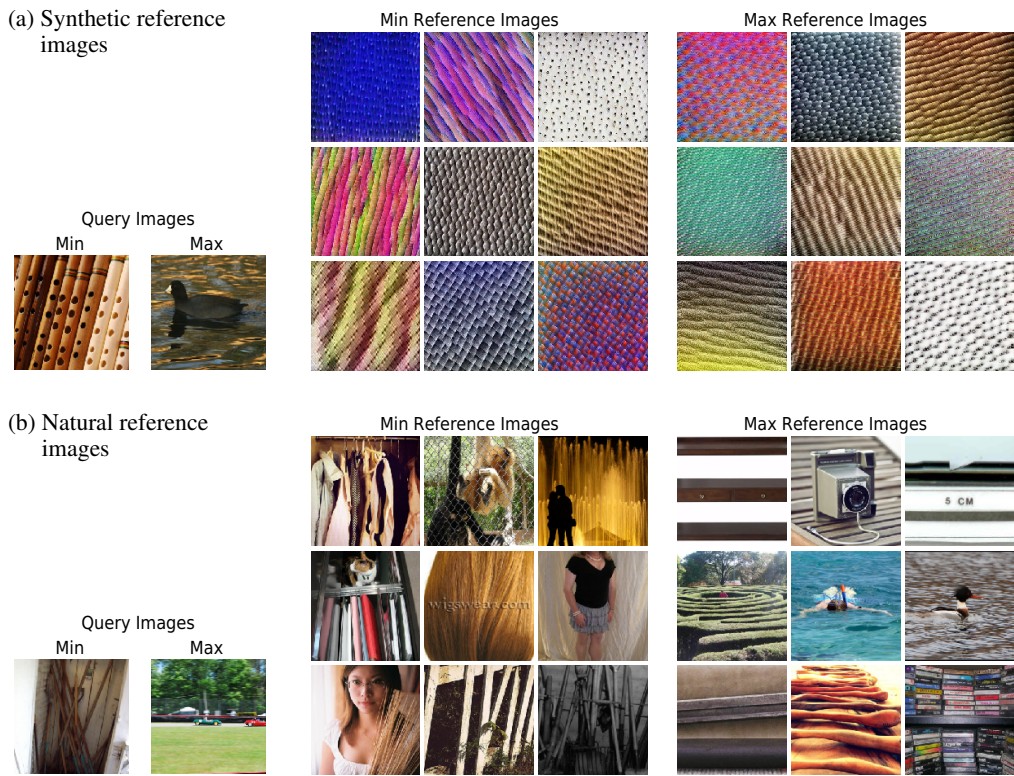

Figure 26: A feature map from a low layer (here: 3a, 3x3) from Experiment I where the feature seems to be a low level cue (horizontal vs. vertical striped) that is surprisingly clear in the natural, but surprisingly unclear in the synthetic reference images. In this specific example, seven (eight) out of ten subjects gave the correct answer for this feature map given synthetic (natural) reference images. The displayed stimuli were shown to participant 1, for stimuli shown to participant 2 and 3, see Supplementary Material Fig. 11.

trials were missed). As this indicates that our data is of high quality, we do not perform the analysis with excluded trials as we expect to find the same results.

---

[8]Baseline condition.

[9]Metrics of explanation quality computed without human judgment are inconclusive and do not correspond to human rankings.

[10]Task has an additional "I don't know"-option for confidence rating.

[11]Comparison is only performed between methods but no absolute measure of interpretability for a method is obtained.

## A.3 Details on Related Work

| Paper | Analyzes Intermediate Features? | Explanation Methods Analyzed | Explanation helpful? | Results Confidence/Trust | |
|---|---|---|---|---|---|
| **Ours** | yes | • Feature Visualization
• natural images[8]
• no explanation[8] | yes | • high variance in confidence ratings
• natural images are more helpful | |
| Biessmann & Refiano (2019) | no | • LRP
• Guided Backprop
• simple gradient[8] | yes | • highest confidence for guided backprop[9] | |
| Chu et al. (2020) | no | • prediction + gradients
• prediction[8]
• no information[8] | no | • faulty explanations do not decrease trust | |
| Shen & Huan (2020) | no | • Extremal Perturb
• GradCAM
• SmoothGrad
• no explanation[8] | no | • - | |
| Jeyakumar et al. (2020) | no | • LIME
• Anchor
• SHAP
• Saliency Maps
• Grad-CAM++
• Ex-Matchina | unclear[11] | • - | |
| Alqaraawi et al. (2020) | no | • LRP
• classification scores
• no explanation[8] | yes | • confidence similar across conditions | |
| Chandra-sekaran et al. (2017) | no | • prediction confidence
• attention maps
• Grad-CAM
• no explanation[8] | no | • - | |
| Schmidt & Biessmann (2019) | no | • LIME
• custom method
• random/no explanation[8] | yes | • humans trust own judgement regardless explanations, except in one condition | |
| Hase & Bansal (2020) | no | • LIME
• Prototype
• Anchor
• Decision Boundary
• combination of all 4 | partly | • high variance in helpfulness
• helpfulness cannot predict user performance | |
| Kumaraku-lasinghe et al. (2020) | no | • LIME | yes | • fairly high trust and reliance | |
| Ribeiro et al. (2018) | no | • LIME
• Anchor
• no explanation[8] | yes | • high confidence for Anchor
• low for LIME & no explanation | |
| Alufaisan et al. (2020) | no | • prediction + Anchor
• prediction[8]
• no information[8] | partly | • explanations do not increase confidence | |
| Ramamurthy et al. (2020) | no | • MAME
• SP-LIME
• Two Step | • unclear[11] | • users can adjust MAME which increased trust | |
| Dieber & Kirrane (2020) | no | • LIME | partly | • - | |
| Dinu et al. (2020) | no | • SHAP
• ridge
• lasso
• random explanation[8] | partly | • no statement on confidence ratings | |

| Paper | Experimental Setup | | | |
| | Dataset | Task | Participants | Collected Data |
|---|---|---|---|---|
| **Ours** | • natural images (ImageNet) | • CNN activation classification | • experts
• laypeople | • decision • confidence
• reaction time
• post-hoc evaluation |
| Biessmann & Refiano (2019) | • face images (Cohn-Kanade) | • 2-way classification[10] | • laypeople | • decision • confidence
• reaction time |
| Chu et al. (2020) | • face images (APPA-REAL) | • age regression | • laypeople | • decision • trust
• reaction time
• post-hoc evaluation |
| Shen & Huan (2020) | • natural images (ImageNet) | • model error identification | • laypeople | • decision |
| Jeyakumar et al. (2020) | • natural images (CIFAR-10)
• text (Sentiment140)
• audio (Speech Commands)
• sensory data (MIT-BIH Arrhythmia) | • preference for one out of two explanation methods | • laypeople | • decision |
| Alqaraawi et al. (2020) | • natural images (Pascal VOC) | • classification | • technical background (neither lay nor expert) | • decision
• confidence
• free answer on features |
| Chandra-sekaran et al. (2017) | • VQA (visualqa.org) | • model error identification
• regression | • laypeople | • decision |
| Schmidt & Biessmann (2019) | • book categories
• Movie reviews (IMDb) | • 9-/2-way classification | • laypeople | • decision
• reaction time
• trust |
| Hase & Bansal (2020) | • movie reviews (Movie Review)
• tabular (Adult) | • 2-way classification | • experts | • decision
• helpfulness rating
• explanation helpfulness |
| Kumaraku-lasinghe et al. (2020) | • tabular (Patient data) | • 2-way classification | • experts | • decision
• feature ranking
• satisfaction
• questionnaire |
| Ribeiro et al. (2018) | • tabular (Adult, rcdv) | • 2-way classification[10]
• VQA | • experts | • decision
• reaction time
• confidence |
| Alufaisan et al. (2020) | • tabular (COMPAS, Census Income) | • 2-way classification | • laypeople | • decision
• confidence
• reaction time |
| Ramamurthy et al. (2020) | • tabular (HELOC, pump failure) | • 2-way classification | • experts
• laypeople | • decision |
| Dieber & Kirrane (2020) | • tabular (Rain in Australia) | • interview | • laypeople
• experts | • how interpretable LIME output is |
| Dinu et al. (2020) | • tabular (Airbnb price listings) | • interview | • laypeople | • decision: which model would perform better in practice
• confidence |

Table 4: Overview of publications that evaluate explanation methods in human experiments. Note that the table already starts on the previous page and that the footnotes are displayed on page 39.

