# OpenReview forum: "Exemplary Natural Images Explain CNN Activations Better than State-of-the-Art Feature Visualization"
_ICLR.cc/2021/Conference — ICLR 2021 Poster_

### Official Review · AnonReviewer3 · 2020-10-24
**A potentially good contribution to the literature**

**Rating:** 6
**Confidence:** 4

**Review:**


The manuscript presents a systematic analysis comparing the effectiveness of using the output (explanations) produced by input reconstruction methods (a.k.a. feature visualization) when compared to exemplar (natural) images for the task of explaining the activation of given layer in a neural network.

While the technical contributions of the manuscript, its observations are of significant relevance for the continuously growing literature related with model explanation/interpretation methods applied to neural networks.

Overall, the presentation of the manuscript is good, and to a good extent its content is clear and easy to follow. A significant set of related literature is referred to.

Reading the manuscript was both enjoyable and insightful.

Regarding the novelty of the analysis, I am not familiar with efforts analyzing model explanation and interpretation algorithms from the human side. To the best of my knowledge, this is one of the first I have read related to input reconstruction methods.

Having said that, my main concerns with the manuscript are the following:

The natural image baseline needs to be explicitly defined.
More specifically, a description on how the natural images used in the experiments are selected is not in place. This is a critical aspect of the whole study. This description is not present in the manuscript, it is hosted in the supplementary material instead. If this manuscript is to be accepted, and to ensure that it is self-contained, the description of this baseline should be in the body of the manuscript.

When comparing experts vs. lay users, experts are considered as individuals with experience with CNNs.
This begs the question, what if you have experts that beyond being familiar with CNNs are also familiar with explanation methods? It would be insightful to further conduct experiments with more specialized users, i.e. users with familiarity on the explanation methods.

There is quite some redundancy in the content presented in Sections 4 and 5, I would encourage the combinations of these two sections in favour of providing more details related to how the natural images baseline is defined.

As admitted by the manuscript, one of its limitation is its limited focus to only considering the feature visualization methods from Olah et al, 2017. Including different examples from other families of model explanation methods would have strengthened the manuscript significantly.

---

> ### Author Response · Authors · 2020-11-24
> **Answer to Reviewer 3 - Part 1(2)**
>
> Dear Reviewer 3,
>
> Thank you for your detailed and thorough review! We are happy to hear that you found our observations “of significant relevance” to the community, and that you found reading the paper was “both enjoyable and insightful.”
>
> We notice you state that our paper would be “one of the first [you] have read related to input reconstruction methods.” Despite our detailed literature search (see Tab. 4 in Sec. A.3) we are not aware of another study that examines human interpretability of this kind of feature visualization method quantitatively. In case we have missed such a study we would be thankful for a pointer to integrate it into our manuscript.
>
>
> Regarding your concerns:
>
> Comment 1: *The natural image baseline needs to be explicitly defined*
>
> We agree that an explicit definition of the natural image baseline is critical for understanding our study and, therefore, extended the last paragraph of Sec.3: We explain that we used “the spatial average of a whole feature map (“channel objective”)” for feature visualizations and provide details on how we selected natural images (“[...] the images of the most extreme activations are sampled, while ensuring that each lay or expert participant sees different query and reference images.”) Additionally, we added a more prominent reference to the appendix which itself hosts a very detailed description (Sec. A.1.2)  of the sampling process for both types of stimuli. In general, our goal was to keep the description in the main text concise to enhance the readability of the paper and provide more details of the sampling process for the interested reader in the appendix.
>
> Comment 2: *What if you have experts that beyond being familiar with CNNs are also familiar with explanation methods?*
>
> Interesting question! To address this, we contacted all the participants who took part in our experiment and asked them to rate how familiar they are with explanation methods, since some of our expert participants had indeed worked with explanation methods before. We asked them to rate their familiarity level from (1) never hear of feature visualizations,(2) hear of/read about feature visualizations, to (3) tried/used/actively engaged with feature visualizations,
> where we considered feature visualizations fairly broadly and also accepted e.g. DeepDream. When we investigate the relationship between familiarity and performance, we do not find a clear trend - performance is similarly high, no matter whether participants had worked extensively with explanation methods before. This analysis is now added to the updated manuscript, where we describe the analysis in Sec. A.2.3 and visualize the results in Figure 20. Additionally, we included an analysis by background (e.g. computer vision/computational neuroscience). While those participants who have a background in neuroscience often see/work with maximally exciting images for monkeys and rodents, we surprisingly found that participants whose area of expertise lies in computer vision/ML related topics perform slightly better. With the caveat in mind that the group sizes of both analyses are fairly small, these results further corroborate our previous finding that the performance of lay and expert participants is similar.
>
>
> Comment 3: *Combining sections in favour of providing more details related to how the natural images baseline is defined.*
>
> We were able to incorporate a more detailed description of the natural image baseline in the main text without having to shorten / combine the Results & Discussion sections. We prefer to keep these separate in order to maintain a distinction between the statistical presentation of results (“results section”) and the more forward-looking interpretation of them, along with how the results relate to the broader context and literature (“discussion section”).

---

> ### Author Response · Authors · 2020-11-24
> **Answer to Reviewer 3 - Part 2(2)**
>
> Comment 4: *Including different examples from other families of model explanation methods would have strengthened the manuscript*
>
> The goal of this study is to present the first framework to quantify the helpfulness of feature visualizations in one specific task (understanding the activations caused by natural images). We decided to focus on a single method (by Olah et al. 2017) and give an in-detail analysis of this method and the psychophysical measurements we collected for this goal. To the best of our knowledge, however, the method by Olah et al. (2017) is by far the most popular method and is the only one used to visualize the selectivity of intermediate feature maps using synthetically optimized stimuli. To make sure we do not give the impression that we evaluated multiple visualization methods, we updated the title (“Exemplary Natural Images Explain CNN Activations Better than State-of-the-Art Feature Visualization”) and adjusted the text of the manuscript (see Sec. 2: removed details on saliency maps, and Sec. 5: mentioned that we only focus on the method from Olah et al. (2017)). Further, we agree that an analysis of and comparison with other feature visualization methods would be of interest for the community. However, we argue that given the high costs for collecting human data in our highly controlled setting as well as the considerable length of the manuscript (39 pages including the appendix), this goes beyond the scope of a single paper and can be better addressed in future work, once this psychophysical evaluation scheme is established.

---

### Official Review · AnonReviewer4 · 2020-10-28
**One reasonable framework for evaluating synthetic feature visualizations**

**Rating:** 5
**Confidence:** 4

**Review:**

This paper asks a simple question: do extreme-activating synthetic images for a CNN unit help a human observer to predict that unit’s response to natural images, compared with maximally/minimally activating natural images. The authors present human observers with images synthesized to maximally or minimally activate a CNN unit, and then ask observers to make a binary choice as to which of two subsequently presented natural images will yield a larger unit response. They find that the synthetic images provide useful information for prediction, but that the benefit is smaller than that provided by simply presenting people with other natural images that maximally or minimally activate a unit.

--

Pros:

The paper presents a simple and reasonable framework for evaluating the utility of synthesized images in allowing human observers to make predictions about how a CNN will behave.

The paper is clear and to-the-point.

The paper examines the utility of synthetic images at multiple stages/layers of a CNN, finding similar results at all stages.

I liked that the authors considered both experts and lay people.

--

Limitations and questions:

Feature visualizations have many potential goals, only one of which is to allow a human observer to predict network behavior for natural images. For example, feature visualizations may provide insights about how the network’s behavior deviates from that of a human observer, by showing that the network relies on features that differ from human-intuitive features. Feature visualizations might also be used to compare the strategies used by two different networks to perform the same task, which might be hard to do using natural images alone. This multiplicity of uses is something which I think should be highlighted more prominently. -- Update: the authors have made minor text changes to note limitations. --

I wondered what the results would look like for a simpler case: a linear filter? In the case of a linear filter, it’s clear that the filter captures everything necessary to predict the response of the unit. But it's not obvious to me how good human observers would be at using the filter to predict how a linear filter would behave in natural images. Would the results be the same if the authors looked at first the layer of the network? If so, does this suggest that humans are just bad at using synthetic stimuli to make predictions in natural images? -- Update: The authors note that feature visualizations are no more helpful for lower layers. This seems consistent with the idea that the limitation is with the human observer as opposed to the stimuli. --

The authors only examine a single kind of feature visualization, which in my opinion limits the overall impact of the paper. For example, I wondered if visualizations that highlight relevant pixels or regions of an image would help observers in making predictions. -- Update: The authors have done more to acknowledge this limitation. --

--

I don’t have a strong recommendation to accept or reject, at the moment. The paper doesn’t seem particularly ground breaking. But it’s solid, and introduces a useful framework for evaluating feature visualizations.

---

> ### Author Response · Authors · 2020-11-24
> **Answer to Reviewer 4**
>
> Dear Reviewer 4,
>
> Thank you for your detailed and constructive review! We are glad to hear you appreciated our “simple and reasonable framework” and that you found our paper “clear”, “solid” and “useful.”
>
> Regarding the “limitations and questions”:
>
> Comment 1: *Feature visualizations have many potential goals, only one of which is to allow a human observer to predict network behavior for natural images*
>
> We agree that feature visualizations can be used in multiple ways, and we adjusted our manuscript at two locations to make this point clearer: In the Discussion & Conclusion section (Sec. 5) - where we already elaborate on the fact that our forward simulation paradigm is only one specific way to measure the informativeness of explanation methods - we now added that this paradigm “does not allow us to make judgments about their helpfulness in other applications such as comparing different CNNs.” Further up, in the paragraph of the first question, we added the aspect that feature visualizations are independent of the “natural image manifold” and therefore reveal “unconstrained features used by a CNN.” We then go on and hypothesize that “ever growing datasets and compute resources” may allow us to discover “similar - if not even better - insights.”
>
> Comment 2: *I wondered what the results would look like for a simpler case: a linear filter?*
>
> This is an interesting question! At the moment, we do not have data on a truly linear filter, as we did not include such a filter in our experiment. Since data collection is expensive and time-consuming, we plan to look into this in the future. From our existing data (see Fig. 22), we observe no evidence that synthetic feature visualizations would provide a better cue for lower than for intermediate or higher layers. As illustrated in Fig. 26, we observe that feature visualizations for lower layers look surprisingly diverse - and hence often confusing. We hypothesize that the diversity aspect in the objective counteracts that a clear feature would be displayed. Above all, we imagine that the performance for feature visualizations of lower layers largely depends on how complex their features would be (Olah et al. (2020) display these very low layer features here: https://distill.pub/2020/circuits/early-vision/)
>
>
> Comment 3: *Only a single kind of feature visualization is examined*
>
> You’re right: We only investigated a single feature visualization method, and we agree that it would be a natural follow-up experiment to investigate other visualization methods. To the best of our knowledge, however, the method by Olah et al. (2017) is by far the most popular method and is the only one used to visualize the selectivity of intermediate feature maps using synthetically optimized stimuli. Additionally, running a careful human evaluation study in a controlled psychophysical lab setup, and adhering to a strict hygiene protocol during a pandemic takes a substantial amount of time (on the timescale of months rather than weeks) and resources. Therefore, we believe that our careful examination of a single method “in depth”, including comparisons between experts/lay people, investigating several layers, and a comprehensive investigation of the different presentation schemes, jointly lay the groundwork for future studies on other visualization methods.
> Nonetheless, we have identified various sentences where our claims were broader than necessary, given that only a single method was investigated in-depth. Consequently, we re-phrased the text at several locations:
>
> - Title: Exemplary Natural Images Explain CNN Activations Better than State-of-the-Art Feature Visualization
> - Abstract:
>     - “The experiment is designed to maximize participants' performance, and is the first to probe intermediate instead of final layer representations.”
>     - “the interpretability of the feature visualizations by Olah et al. (2017)”
>     - “Synthetic images from a popular feature visualization method are significantly less…”
> - Sec. 2, Related Work:
>     - remove details on saliency maps as to not falsely imply that we would cover anything else but the feature visualization method by Olah et al. (2017)
>     - Add to the last paragraph that we only use images “generated with the method of Olah et al. (2017)”
> - Sec. 5, Discussion & Conclusion:
>     - Add to the first paragraph that we only use synthetic feature visualizations “by Olah et al. (2017)”

---

### Official Review · AnonReviewer2 · 2020-10-28
**Interesting results, study has a natural bias that may constrain broadness of claims**

**Rating:** 8
**Confidence:** 5

**Review:**

##Updated Review##

I'd like to thank the authors for addressing each of the points I made in my review and taking the time to include material that answers many of the questions I had.  This paper is in my mind a novel and interesting submission and a clear accept.

# Main Idea
The main idea is to study how well extremely activating images help humans to predict CNN activations.  The authors do so by comparing extremely activating images with exemplary natural images that also strongly activate a specific feature map (and use a psychophysics test to see which helps the human better).

# Some quick thoughts
The authors point that many visualization methods blend response maximization with human-define regularization methods which are essentially artistic choices meant to make the image less noisy is well taken and correct.  And these regularizations impute their own biases on the resulting images which may make them less informative.

It is also well taken that units may be highly activated by more than one semantic concept, or active in combination with other units (which may convey more information than selectively maximizing for the single neuron’s activation).

While the authors points are well taken, they do sort of sweep aside one aspect of feature visualization that is not captured by natural images.  That is that feature visualization shows how and where a particular neuron’s response is not constrained to natural image statistics (and is potentially vulnerable to image manipulations).  This isn’t revealed in their psychophysics experiment because they test on the prediction of typical natural images only, which won’t contain these deviations from the natural image manifold.  The disconnect in predictability for future natural images actually reveals information about the networks (not necessarily a failure of the visualization technique!)

If the only goal of explainable AI is to help us predict what natural image a neuron in a mid level feature map will respond to strongly, then this study shows that using natural images helps with that.  However, if it is also to elucidate something about the working of the networks beyond constraint to the manifold of natural images, then the bias in this study will mislead us about which method is more useful.

The experiment in Figure 5C (showing different numbers of samples of maximally exciting images of each class) reveals something interesting here.  How does the diversity regularization utilized in this section effect variance in the feature visualization images when compared to the different samples of naturally exciting images in this set?

# Weaknesses:

Lots of broad claims made about feature visualization vs natural images but really only compare to one specific method of creating synthetic images and at a particular setting of extreme activation.

Figure 4 would be easier to digest if the data were on a single common axis and the colors of the bar separated between synthetic and natural (as such it is a bit difficult to compare layer to layer. There is interesting information in this correspondence!  For example, why do layers 4b and layer 5b show similar performance for both natural and feature visualization images?

Figure 10 in the supplementary also shows very interesting relationship between the response strength between the natural images and the synthetic images.  It seems like attempting to balance the response level between image types would be a good control for future experiments.  This also highlights a point raised early, that the manifold of natural images imposes constraints on how exactly these images can be configured and thus actually can’t show us certain response characteristics of these neurons).  This gets into what exactly you are using these methods for, the predict natural images they may respond to or a deeper analysis of their response characteristics.

---

> ### Author Response · Authors · 2020-11-24
> **Answer to Reviewer 2**
>
> Dear Reviewer 2,
>
> Thank you for your detailed and constructive review! We are happy to hear that you found our results “interesting” and that you consider several pieces of greater context “well taken”.
>
> We will answer your points by topic in chronological order:
>
> Comment 1: *Feature visualization shows how and where a particular neuron’s response is not constrained to natural image statistics*
>
> We agree that an advantage of feature visualizations is their independence of the natural image manifold and adjusted the manuscript accordingly (see Sec. 5 Discussion & Conclusion, paragraph 2). On a different note, we argue that if feature visualizations are indeed able to better work out the features to which a feature map responds, then we should expect high performance also if tested on natural images, which arguably is the most relevant data on which we aim to understand the model responses.
>
>
> Comment 2: *The disconnect in predictability for future natural images actually reveals information about the networks (not necessarily a failure of the visualization technique!)*
>
> We share the opinion that feature visualizations can be used in multiple ways. In our caveats paragraph of Sec. 5 Discussion & Conclusion, we therefore extend our statement that “the forward simulation paradigm is only one specific way to measure the informativeness of explanation methods” by “[this paradigm] does not allow us to make judgments about their helpfulness in other applications such as comparing different CNNs”. In the bigger picture, we want to state that we agree with Leavitt and Morcos (2020) in that explanation methods (including feature visualizations) suffer from an “over-reliance” on intuition and should more extensively be tested against falsifiable hypotheses. We hope our work can be seen as one step in that direction.
>
>
> Comment 3: *How does the diversity regularization utilized in this section effect variance in the feature visualization images?*
>
> To illustrate the diversity regularization on feature visualizations, we added both natural and synthetic images for a selection of feature maps in a new section in the Appendix A.2.7 and in the Supplementary Material. We chose the feature maps such that there are examples of various performance levels for natural/synthetic reference images (see Fig. 23 to 26). As we describe in the text of Sec. A.2.7, our impression is that (1) easy feature maps seem to have clearly identifiable features, (2) confusion occurs when reference images diverge, yet (3) high variance in natural images of low layer feature maps can be beneficial to abstractly convey a low level concept (see Fig. 25b).
>
> Comment 4: *Broad claims made about feature visualization vs natural images but really only compare to one specific method of creating synthetic images*
>
> To the best of our knowledge, the method by Olah et al. (2017) is the most popular method and the only one widely used to visualize the selectivity of intermediate feature maps using synthetically optimized stimuli. It is commonly just referred to as “feature visualisation” with no other denomination. We nonetheless agree that the word “feature visualisation” suggests a more broader family of methods, and we hence adjusted the title$^1$ and several parts of the text$^2$ accordingly to clarify the scope of this paper.
>
> $^1$ “Exemplary Natural Images Explain CNN Activations Better than State-of-the-Art Feature Visualization”
>
> $^2$ All sections now explicitly clarify that we investigated the feature visualization method by Olah et al. (2017)
>
>
>
>
> Comment 5: *Figure 4 would be easier to digest if the data were on a single common axis*
>
> Following your suggestions, we updated Figure 4 to show the bars for natural and synthetic images in an alternating fashion to make comparing the two easier. Regarding similar performance in certain layers, we want to point out that that data from many more participants would be required to perform a solid by-layer analysis. From these data we draw the broader conclusion that that performances are roughly similarly high for all layers / branches.
>
> Comment 6: *Figure 10 shows very interesting relationship b/w the response strength b/w the natural images and the synthetic images*
>
> We present our findings of balancing activation levels between synthetic and natural images in the Appendix Sec. A.1.3. Specifically, we generated synthetic images that match the activation levels of natural examples by stopping the optimization process early. The final activation levels are shown in Fig.10b. Example images in our new Fig. 11 show that  they are visually very different from the “normal” feature visualizations and that running the optimization process longer is a reasonable choice - even when the price is that activation levels differ largely.

---

### Official Review · AnonReviewer1 · 2020-11-09

**Rating:** 7
**Confidence:** 3

**Review:**

--- Summary ---

This paper focuses on feature visualizations that generates maximally activating images for a given hidden node to understand inner workings of CNNs. They compare the informativeness of these images compared to natural images that also strongly activate the specified hidden node, and find that natural images help human better to answer which other test natural images are also maximally activating.

--- Pros ---

1. The writing is crisp and excellent: the flow in the intro is easy to follow, the related work is thorough, and the figures are designed very well.

2. The experimental design is clear and reasonable. The statistical comparisons are rigorous and the visualizations are clear.

3. I like the subjective score section: the qualitative conversations are intersting to read.

--- Comments ---

1. Are there some examples that natural images can not help the participant predict while synthetic images can, and vice versa? It might be interesting to see in what circumstances which methods help.

2. The conclusion is not surprising given the task is designed to find other test natural images. But this paper does highlight the shortcomings of synthetic feature visualizations that are not easy for human to understand.

--- Overall evaluation ---

The writing is great and clear. Although I find the conclusion is not very surprising, I still enjoy reading this paper throughout its rigorous experimental setups like experts v.s. laymen, hand-picked v.s. random images, number of images presented, and subjective scores etc.

---

> ### Author Response · Authors · 2020-11-24
> **Answer to Reviewer 1**
>
> Dear Reviewer 1,
>
> Thank you for your positive and constructive review! We are happy you liked our “rigorous experimental setups”, our “crisp and excellent” writing, as well as our “rigorous” statistical comparisons and “clear visualizations”.
>
> Regarding your comments:
>
> Comment 1: *Are there some examples that natural images can not help the participant predict while synthetic images can, and vice versa?*
>
> To address this question, we added a paragraph at the end of Sec. 4.6 with an analysis of individual feature maps from Experiment I and a detailed section in Appendix A.2.7 “By-Feature-Map Analysis”. We indeed found two feature maps where the performance was low (only four out of ten subjects gave the correct answer) for one reference type, but high (nine out of ten correct answers) for the other one (see Fig. 24). For each of these feature maps, we notice that the images of the more difficult reference condition (respectively the synthetic or natural images) are very diverse.
> To put this analysis into some more context, we precede the difficult feature maps with a plot showing the number of correct answers for each individual feature map (Fig. 22), analyze and show easy feature maps (Fig. 23), and also share our impressions on what aspects make feature visualizations difficult: They “seem to have diverse reference images, features that do not correspond to human concepts, or contain conflicting information as to which commonalities between query and reference images matter more.” (Sec. 4.6). At the end of A.2.7, we also address the question from Reviewer 4 regarding linear filters. In addition, we added additional examples for different feature maps in the Supplementary Material.
>
> Comment 2: *The conclusion is not surprising given the task is designed to find other test natural images. But this paper does highlight the shortcomings of synthetic feature visualizations [...]*
>
> We understand the perspective that our task setup would contain a bias. However, it’s not entirely clear to us as to whether feature visualisations really are at a disadvantage in our setup: The method is not constrained by the natural image manifold and may thus work out more clearly to what the feature maps respond to, which in turn should help in our task. As we point out in the discussion section, the kind of images that explanation methods will be applied to in the real world will also be natural. Therefore, we consider our choice of testing humans on natural images as the most reasonable starting point. Regarding our conclusion: this paper is (to our knowledge) the first to quantify the helpfulness of synthetic feature visualisations for humans. We show that a synthetic feature visualization method indeed provides useful information: with a performance of 82%±4%, it performs much better than chance (50%). The surprising finding is that at least this method performs more poorly than the natural reference image baseline. We believe that our paper offers a starting point for quantitatively evaluating explainability methods, and that our current result offers useful insights in how to improve future explainability methods (see the new discussion and analysis added in response to your suggestion above).

---

### Decision · Program_Chairs · 2021-01-07
**Final Decision**

**Decision:**

Accept (Poster)

**Comment:**

This paper asks a simple question: Do extreme-activating synthetic images for a CNN unit help a human observer to predict that unit’s response to natural images, compared with maximally/minimally activating natural images. They authors conducted well-designed human studies and found that the synthetic images provide useful information for prediction, but that the benefit is smaller than that provided by simply presenting people with other natural images that maximally or minimally activate a unit.

The paper provides one reasonable metric for evaluating feature visualizations. Feature visualizations are widely used, but there are very few objective metrics for evaluating them. This methodological contribution is the main contribution of this work and could be impactful. Although the conclusion is not very surprising, the paper makes a potentially good contribution to the literature.